# Auditory cortical micro-networks show differential connectivity during voice and speech processing in humans

Florence Steiner [1,2,4], Marine Bobin [1,2,4] & Sascha Frühholz [1,2,3 ✉]

The temporal voice areas (TVAs) in bilateral auditory cortex (AC) appear specialized for voice processing. Previous research assumed a uniform functional profile for the TVAs which are broadly spread along the bilateral AC. Alternatively, the TVAs might comprise separate AC nodes controlling differential neural functions for voice and speech decoding, organized as local micro-circuits. To investigate micro-circuits, we modeled the directional connectivity between TVA nodes during voice processing in humans while acquiring brain activity using neuroimaging. Results show several bilateral AC nodes for general voice decoding (speech and non-speech voices) and for speech decoding in particular. Furthermore, non-hierarchical and differential bilateral AC networks manifest distinct excitatory and inhibitory pathways for voice and speech processing. Finally, while voice and speech processing seem to have distinctive but integrated neural circuits in the left AC, the right AC reveals disintegrated neural circuits for both sounds. Altogether, we demonstrate a functional heterogeneity in the TVAs for voice decoding based on local micro-circuits.

[1] Cognitive and Affective Neuroscience Unit, University of Zurich, Zurich, Switzerland. [2] Neuroscience Center Zurich, University of Zurich and ETH Zurich, Zurich, Switzerland. [3] Department of Psychology, University of Oslo, Oslo, Norway. [4] These authors contributed equally: Florence Steiner, Marine Bobin. ✉email: sascha.fruehholz@uzh.ch

The temporal voice area (TVA) is a neural cluster specialized for voice processing in the auditory cortex (AC)[1]. The cortically extended TVA includes three bilaterally symmetric patches located on the caudal-to-rostral axis in posterior (pST), mid (mST), and anterior superior temporal cortex (aST)[1,2], pointing to a diverse collection of TVAs. The TVA and especially its patches have been found to rather uniformly respond to voices on a perceptual level beyond the decoding of basic acoustic features[3,4]. By this notion of a uniform response profile of the TVA and its subpatches, we refer to the observation that previous studies did not directly test a potential functional difference between subareas of the TVA or its patches[1]. Previous studies only reported some posthoc and rather indirect evidence[5] as well as theoretical explanations[2,6,7]. This lack of investigations left some uncertainty regarding the differential involvement of the voice patches in voice recognition processes. However, based on these previous reports, there might be evidence that some subparts of the TVA decode certain types of voice information[7], such as voice identity and speech decoding in anterior TVA[8–10] and voice-specific acoustical processing in mid-TVA[11–13]. Nonetheless, a clearer and direct functional description of TVA subareas for generically discriminating voice signals from other auditory signals at the functional level of auditory object discriminations and classifications is missing.

Given these previous reports of a functional homogeneity for voice processing in the TVA, including the frequently proposed voice selectivity[14], the notion of a spatially extended organization of bilateral TVAs including multiple voice subpatches seems a little bit surprising. Considering the differential working principles of AC subregions underlying the TVA[15], this uniformity might be rather unlikely. We here accordingly tested if the TVA is instead composed of a local and differential AC network that functionally shifts sound information across TVA subregions to discriminate various nonverbal (non-speech) and speech-based voice sounds from other sounds. We used functional neuroimaging in 52 human participants while they listened to various voice (speech, nonverbal) and non-voice sounds (animals, natural—such as the sound of rain, and artificial—such as the sound of a car).

We tested two main questions in our study. Our first question concerned the notion of rather integrated or rather separated neural pathways for voice signal processing and voice signal differentiation in the AC. By integrated neural pathways, we refer to the potential observation that different types of voice signals (voice signals in general, speech signals) are processed along similar neural pathways in the AC, and also that this processing integrates the distributed information decoding across different local AC processing nodes. Contrarily, separated neural pathways would show disintegrated processing of various voice signals, separated across different neural AC nodes. More specifically, we expected that if the neural processing of voice and speech signals is integrated, activation patterns observed for these two types of voice signals would co-localize in similar brain areas of higher-order AC since separate studies have shown similar peak activity locations for voice processing[1,16] and for speech recognition[17,18] from anterior to posterior ST. However, there are also indications of a spatial separation for the neural processing of voice and speech signals[19], pointing to some functional dis-integration of both processes. The same reasoning would apply to the expected neural network underlying voice and speech processing in terms of neural (dis-)integration. If speech processing (speech as a specific voice signal) would depend on the more general voice processing (voice signals as a general category), we would expect that neural speech processing nodes would integrate with and hierarchically follow the neural voice processing nodes in terms of the neural network architecture[20]. However, voice processing could also be neurally disintegrated from neural speech processing, since voice signals are also used to decode socially relevant information apart from speech information[21,22].

Our second question concerned whether cortical voice processing follows documented functional hierarchies in the AC for auditory object processing[20]. A typical processing hierarchy in the AC includes an information flow from primary, secondary, to higher-level AC regions, which has been largely demonstrated for sound feature processing and the processing of simple auditory objects[20]. Common models for the neural processing of auditory objects and communications signals also suggest an anterior-oriented and posterior-oriented gradient for voice and speech signal analysis in higher-order AC[21,23], referring to the origins of a ventral and dorsal processing stream for a detailed communication signal analysis. The neural treatment of rather complex and socially relevant voice sounds might however also include a non-hierarchical AC processing, which dynamically shifts relevant information between low-order and higher-order AC regions[5,24] for sound analysis as well as voice detection and voice type discrimination, respectively. Concerning this question of a (non-)hierarchical organization of the AC micro-networks for voice and speech processing, we expected that the neural network would follow a processing stream from primary/secondary AC to mST, and from mST to either aST (ventral stream) or pST (dorsal stream) in case of a strongly hierarchical organization. In the case of a non-hierarchical organization, we especially expected feedforward and/or backward projections between low- and higher-order AC as well as a neural co-dependence of voice and speech processing (i.e., neural nodes for voice and speech processing influence each other) rather than a strict neural hierarchy (i.e., neural speech processing is dependent on voice processing nodes).

To answer these questions, we used dynamic causal modeling (DCM) to determine the effective neural network connectivity of fMRI data from the human AC during voice processing. DCMs are generative models of neural connectivity in a Bayesian statistical framework that allow modeling the directional connectivity between brain regions based on experimental conditions that influence causal interactions in a defined neural network[25]. Using DCM, we tested networks with three major properties: (a) certain sound conditions provide input to the neural network (i.e., they drive neural activity in network nodes), (b) the network has general effective connectivity between regions that are independent of sound conditions (i.e., intrinsic and extrinsic connectivity reflecting within-node and between-node coupling), and (c) certain sound conditions can modulate the connectivity between network nodes (i.e., modulate within-node and between-node coupling). Previous studies attempted to determine the neural network for voice processing but focused on a non-directional network analysis (i.e., without modeling and determining the direction of the connections in an empirical Bayes framework)[5] and/or by only focusing on large-scale inter-lobe and inter-hemispheric neural networks[26,27].

For an AC neural network analysis of voice processing, we therefore entered left and right neural nodes for voice and speech processing into a DCM analysis. We built the DCM models based on anatomical AC connectivity patterns between low-order and higher-order AC nodes[28–30], including the AC nodes that were specific to nonverbal[31] and speech-based voice processing[32] as the two common modes for voice signal analysis. We tested left and right AC functional connectivity models on a local micro-network level with the following constraints: (a) We allowed bidirectional connections between neighboring nodes in the AC, given that AC regions predominantly communicate with neural nodes in short-range connections[20,23]; (b) driving inputs to neural nodes were defined by dominant experimental conditions

into each node; and (c) modulation of connections between nodes by specific voice and speech sound condition.

These general neural network architectures for the left and right AC were then entered as full connectivity models into a parametric empirical bayes (PEB)[33] procedure to iteratively eliminate uninformative network parameters. The PEB procedure includes three important subcomponents in a Bayes framework to search the connectivity model space and to estimate network parameters: First, it simultaneously models and estimates parameters on the participant and group data level; second, using a Bayesian model reduction (BMR) approach the PEB approach prunes away rather unimportant and non-consistent network parameters as quantified by their posterior probability; and third, using a Bayesian model averaging (BMA) approach, the final model and its network parameters (i.e., posterior probabilities) result from the weighted averaging of these parameters, and thus takes into account all estimated models from the BMR approach leading to valid estimation and representation of neural network properties. Based on this DCM-PEB modeling approach, the resulting connectivity patterns were expected to represent underlying functional pathways for voice and speech processing in local AC networks in the left and the right hemisphere. We tested the AC neural networks for voice processing separately in the left and the right AC because we were mainly interested in determining the local AC micro-network rather than inter-lobe and inter-hemispheric large-scale networks[26,27].

## Results
Before modeling the neural AC connectivity, we first defined the left and right AC nodes that responded to voice sound processing in general and to speech sound in specific. Contrasting neural activity for all voices against all non-voice sounds ($n = 52$ participants), we found broadly extended activity in both left and right AC that commonly defines bilateral TVAs[1,34]. This activity showed a large spatial extent with local activity peaks distributed from anterior to posterior ST, including broad coverage of the higher-order auditory cortical region Te3 (Fig. 1a)[35]. Similar to previous observations[1,5], we found three peaks in the left AC (aST, pST, posterior superior temporal sulcus (pSTS)) and three peaks in the right AC (aST, mST, pST)[36] (Fig. 1a, Table 1a). Activity in bilateral aST/pST was common to both speech-based and nonverbal voice processing (Table 1b, c), while activity in left pSTS and bilateral mST seemed specific for speech processing as a specific type of voice sounds (Fig. 1b, lower panel; Table 1d). For each of those peak locations we created an ROI (a sphere of 3 mm radius around the peak) to be used in the DCM analysis. Figure 1d shows the beta estimates (i.e., the level of activation) in each condition for each of the ROIs.

Using this observed specialization in ST/STS subregions for voice and speech processing, we defined differential neural networks for left and right AC using DCM (Fig. 2). We modeled left and right AC networks separately[24,37], given that we were primarily interested in the local AC micro-networks and given that we aimed at keeping the entire model space in an appropriate range. Our DCM models included three sets of important network parameters: (a) We included driving inputs (C matrix) to each neural node given the specific experimental condition that was driving activity in this region (voice, speech, or all sounds); driving inputs were time-series of neural node activations that were mean-centered prior to entering them in the DCM models; (b) effective connectivity within (intrinsic) and between nodes (extrinsic) was included as coupling parameters between neighboring nodes (A matrix); the resulting A matrix parameters reflect the mean connection weights across all experimental conditions[33]; and (c) certain experimental conditions were allowed to linearly modulate the connections between neural nodes (B matrix); the modulatory condition was chosen based on the specific condition driving the activity of the node that was the origin of the connection (e.g., the connection from left mST to

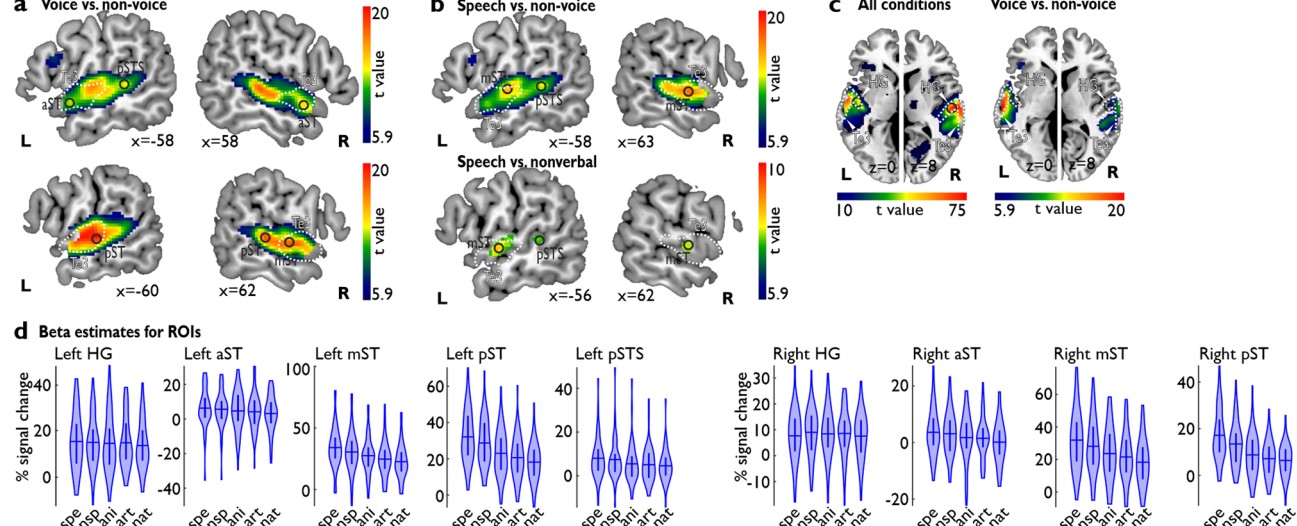

**Fig. 1 Neural AC activity for voice and speech processing.** Contrast images were thresholded at $p < 0.05$ including a voxel-wise FWE correction ($n = 52$ human participants). White dashed outline represents the auditory region Te3. **a** TVA in bilateral AC for contrasting voice against non-voice sounds. Four peak locations were found in left AC, with three peaks located within the auditory region Te3 (aST, mST, pST) and one posterior to Te3 in posterior superior temporal sulcus (pSTS). Two peaks in the right AC were located inside Te3 (aST, mST) and one posterior to Te3 in pST. **b** Contrasting speech against non-voice sounds [Speech vs. non.voice] revealed two left AC peaks (mST, pSTS) and one right peak (mST); these peak activations were confirmed when specifically contrasting speech against nonverbal voices [speech vs. nonverbal]. **c** AC activity for all five conditions compared against baseline [all conditions], with specific peaks in bilateral HG that were also located inside the TVA [voice vs. non-voice]. **d** Violin plots for the percent signal change quantified from beta estimates for all five conditions (*spe* speech, *nsp* nonspeech/nonverbal, *ani* animal, *art* artificial, *nat* natural sounds) for all ROIs (built as a sphere of a 3 mm radius around peak locations); horizontal bar indicates the mean, the inner vertical indicates the first to third quantile of the data distribution.

**Table 1 Neuronal peak activations.**

| Region | T value | MNI | | |
|---|---|---|---|---|
| | | x | y | z |
| (a) *Voice>non-voice* | | | | |
| L aST* | 12.10 | −58 | 4 | −8 |
| L pST* | 22.00 | −62 | −20 | 0 |
| L pSTS | 11.19 | −58 | −36 | 6 |
| R aST* | 14.66 | 58 | 4 | −8 |
| R mST | 18.34 | 62 | −8 | −4 |
| R pST* | 17.51 | 60 | −26 | 2 |
| (b) *Speech>non-voice* | | | | |
| L aST | 7.23 | −54 | 10 | −14 |
| L mST | 18.71 | −60 | −8 | −2 |
| L pST | 20.32 | −62 | −20 | 0 |
| L pSTS | 12.20 | −58 | −36 | 4 |
| R aST | 12.61 | 58 | 4 | −8 |
| R mST | 16.57 | 62 | −10 | −2 |
| R pST | 15.16 | 58 | −26 | 0 |
| (c) *Nonverbal>non-voice* | | | | |
| L aST | 8.41 | −58 | 4 | −8 |
| L pST | 15.10 | −62 | −20 | 0 |
| R aST | 11.17 | 58 | 4 | −8 |
| R pST | 13.48 | 60 | −26 | 2 |
| (d) *Speech>nonverbal* | | | | |
| L mST* | 8.23 | −60 | −8 | −2 |
| L pSTS* | 5.82 | −56 | −36 | 4 |
| R mST* | 5.48 | 62 | −12 | −2 |
| (e) *Nonverbal>speech* | | | | |
| – | | | | |
| (f) *F contrasts (5 conditions)* | | | | |
| L HG* | 75.76 | −44 | −18 | 0 |
| R HG* | 99.41 | 53 | −16 | 8 |

MNI coordinates of functional peak activations for contrasts between the five experimental conditions ($n = 52$ human participants, degrees of freedom (df) = [1, 204]). Contrast images were thresholded at $p < 0.05$ including a voxel-wise FWE correction. Peak locations marked with * were entered into the DCM analysis.
*aST* anterior superior temporal cortex, *mST* mid superior temporal cortex, *pST* posterior superior temporal cortex, *pSTS* posterior superior temporal sulcus, *HG* Heschl's gyrus.

aST was allowed to be modulated by the *speech* condition, as mST was responding with higher activity to speech sounds). Besides regions in the ST, these neural networks finally also included nodes in bilateral Heschl's gyrus (HG) that were sensitive to any incoming sound (Fig. 1c; Table 1f). The sensitivity to any incoming sound is not surprising, given that large parts of the HG, including the parts activated here, belong to the primary AC (cortical region Te1)[35,38], which acts as the low-level processing unit of all kinds of acoustic stimuli. Specifically, also the TVA often includes HG activity[1], which seems to serve as a basic acoustic processing node in the TVA network. More importantly, the HG was included in the DCM models in order to test for a hierarchical or non-hierarchical network architecture for voice and speech processing[20].

The full left and right DCM models (Fig. 2) were then entered into a hierarchical PEB analysis ($n = 52$ participants) on the A and B matrix parameters[33]. This resulted in a left effective neural network (i.e., mean functional connectivity independent of experimental conditions, A matrix) with positive and excitatory bidirectional connections between HG and ST nodes for voice processing and with positive forward and negative backward connections from HG to ST speech processing nodes (Fig. 3a, left panel; Table 2). There were also positive connections between voice and speech processing nodes in ST. Some of these connections were additionally modulated by the experimental conditions as evidenced by significant parameters of the B matrix (Fig. 3b, left panel; Table 2). First, the aST and pSTS seemed relatively independent nodes for voice and speech processing, respectively, given that they showed only minor and negative modulation of forward connections originating from these nodes. Second, there was a specific integrated HG-ST network, which included a sub-branch of HG-pST connectivity where

modulations of connections supported general voice processing, and a HG-mST sub-branch with modulated connections for specific speech processing. Both mST and pST had modulated inter-connections, such that general voice processing might support specific speech processing and vice versa.

Compared to the left neural AC network, the network architecture for voice and speech processing in the right hemisphere showed both similarities and differences (Fig. 3, right panels). In terms of basic network connectivity (A matrix) (Fig. 3a, right panel; Table 3), as in the left hemisphere, the right ST voice processing nodes showed positive and excitatory bidirectional connections from and to HG. However, the mST, as the sole speech processing node, only showed a positive connection to HG but received no input from it. Yet, in addition to the unidirectional positive connections between the ST nodes in the left AC, here the neighboring ST nodes were all bidirectionally connected. While the mST showed positive connections to both voice processing nodes, the aST had a negative and the pST a positive connection to the mST. Thus, there were slightly differential network effects in the anterior and posterior AC in the right brain, which was also confirmed by the condition-specific modulation of connections in the right AC (B matrix) (Fig. 3b, right panel; Table 3). The pST showed a negative and thus inhibitory modulation from HG and the aST negative modulation to HG. The aST showed very similar independence in both hemispheres. Unlike the positive integration of neural processing in pST and mST in the left AC, in the right AC, we found that pST and mST were mostly negatively integrated for voice and speech processing, including the HG as a low-level AC region.

## Discussion

In the present study, we aimed to investigate if the bilateral TVAs, which were previously found in humans[1] and nonhuman primates[39], consist of local and differential AC neural networks that functionally shift sound information across TVA subregions to discriminate various nonverbal and speech-based voice signals from other sounds. We accordingly used functional neuroimaging in humans to determine local cortical peaks in left and right AC that responded with higher activity to voice than to non-voice sounds. Based on the GLM approach to contrast functional brain activity between conditions (Fig. 1), we found three peaks in the left AC (aST, pST, pSTS) and three peaks in the right AC (aST, mST, pST)[36] that seemed sensitive to voice sounds in our experiment. These activation patterns are similar to previous reports[1,5], such that all peak coordinates were completely overlapping with a previously defined TVA probability map[1] (https://neurovault.org/images/106/). Bilateral aST, mST and left pST were located inside the higher-order auditory region Te3, which seems a region that is central to neural dynamics voice processing in the AC[16,27]. Finding this distribution of individual peaks in the AC from the anterior to the posterior end of ST is a first indication of potential different functional roles of these peaks during voice signal processing[23,40,41]. This distribution of voice processing peaks also points to a potential local neural network, which we aimed to define in both the left and the right AC.

Although there are some limitations in the analogies described between human and nonhuman primates voice-specific areas (see also Bodin and Belin[7]), the existence of voice patches in monkeys similar to humans was partially confirmed through fMRI measurements[39,42] and single-unit recordings[4]. In the different studies, voice-sensitive patches (i.e., cortical activation occurring for conspecific-voice processing) have been reported along the ST (particularly the anterior part[42]) up to the STS, also with a divergence between hemispheres and interesting variability in the results[43]. While at a different extent, such observations suggest

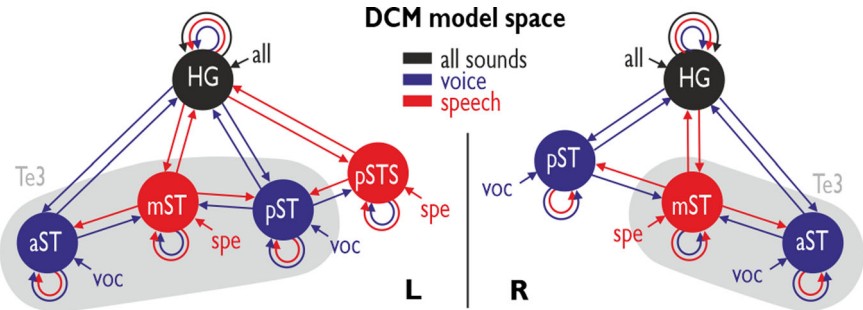

**Fig. 2 DCM model space.** Model space for the DCM analysis including three conditions (all sounds, voice sounds, speech sounds). The full model included driving inputs to each node from different conditions (arrows to nodes), intrinsic connections for each node (circular arrows), bidirectional extrinsic connections between neighboring nodes (straight arrows), and modulations of connections by conditions (color of arrows). Gray shading indicates the auditory area Te3. Black regions were defined based on their activity to all sounds, blue regions were defined based on their activity in the [voice>non-voice] contrasts, and red region were defined by their activity in the [speech>nonverbal] contrast.

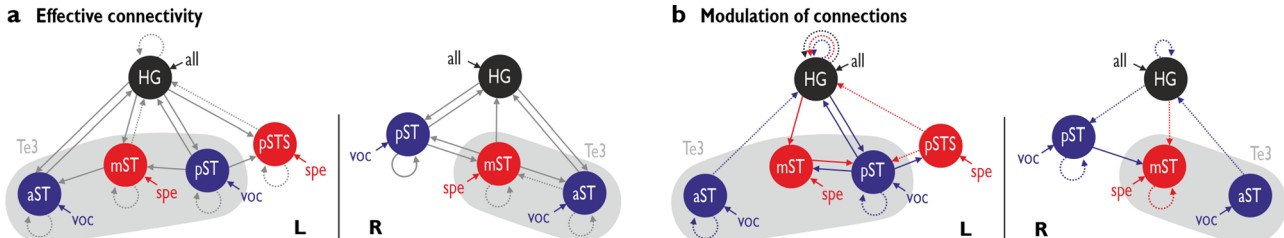

**Fig. 3 DCM effective neural network modeling.** Significant network effects (posterior probability > 0.99) in bilateral AC after a parametric empirical Bayes (PEB)-based model reduction from the full model; sample of $n = 52$ human participants. **a** Neural networks based on the effective connectivity parameters and **b** modulation of connections. Bold lines for positive effects, dotted lines for negative effects. Gray shading indicates the auditory area Te3.

### Table 2 Results of the PEB analysis for the left AC network.

| | HG | aST | mST | pST | pSTS |
|---|---|---|---|---|---|
| *A matrix (intrinsic and extrinsic fixed connections)* | | | | | |
| HG | −0.43 (1.00) | 0.38 (1.00) | −0.19 (1.00) | 0.70 (1.00) | −0.50 (1.00) |
| aST | 0.11 (1.00) | −0.23 (1.00) | 0.37 (1.00) | – | – |
| mST | 0.11 (1.00) | 0.02 (0.62) | −0.43 (1.00) | 0.24 (1.00) | – |
| pST | 0.07 (1.00) | – | 0.00 (0.00) | −0.68 (1.00) | 0.00 (0.00) |
| pSTS | 0.13 (1.00) | – | – | 0.42 (1.00) | −0.39 (1.00) |
| *B matrix (all trials)* | | | | | |
| HG | −2.84 (1.00) | – | – | – | – |
| *B matrix (voice trials)* | | | | | |
| HG | −1.83 (1.00) | −1.59 (1.00) | – | 2.24 (1.00) | – |
| aST | 0.29 (0.96) | −1.07 (1.00) | – | – | – |
| mST | – | −0.37 (0.98) | 0.70 (0.88) | 1.78 (1.00) | – |
| pST | 0.49 (1.00) | – | – | −1.17 (1.00) | – |
| pSTS | – | – | – | 1.13 (1.00) | 0.59 (0.95) |
| *B matrix (speech trials)* | | | | | |
| HG | −1.70 (1.00) | – | 0.00 (0.00) | – | −1.06 (1.00) |
| aST | – | 0.00 (0.00) | 0.00 (0.00) | – | – |
| mST | 1.03 (1.00) | – | 0.72 (0.91) | – | – |
| pST | – | – | 0.94 (1.00) | 0.00 (0.00) | −0.72 (1.00) |
| pSTS | 0.26 (0.92) | – | – | – | 0.00 (0.00) |

Expected posterior parameters (posterior probabilities in brackets) for the A matrix parameters (intrinsic and extrinsic fixed connections) and the B matrix parameters (modulation of connections); sample of $n = 52$ human participants. The matrix is organized such that the origin of connections is represented by the columns, and the target of connections is represented by the lines. *aST* anterior superior temporal cortex, *mST* mid superior temporal cortex, *pST* posterior superior temporal cortex, *pSTS* posterior superior temporal sulcus, *HG* Heschl's gyrus.

that micro-networks specialized for vocalization processing might already be latent in nonhuman primates.

In our study, we included both voice and non-voice sounds to determine the TVA. The voice sounds were comprised of speech and nonverbal (non-speech) stimuli. Speech[32] and nonverbal sounds[31] are both of vocal nature, but they might receive some differential processing dynamics in the AC. In our study using the GLM approach (Fig. 1), we could separate local peaks in the AC that were either sensitive to any voice sounds (speech, nonverbal) or to speech sounds specifically. Unlike the common notion that speech processing is usually accomplished rather down-stream of the auditory processing hierarchy, towards higher-order AC regions at the anterior and posterior ends of the ST[23,44], we found that speech processing units in our study are cortically surrounded by voice processing units, with the exception of the left pSTS. These data thus seem to suggest that speech processing nodes are spatially integrated with voice processing nodes in a local and probably non-hierarchical micro-network. However, we

**Table 3 Results of the PEB analysis for the right AC network.**

| | HG | aST | mST | pST |
|---|---|---|---|---|
| *A matrix (intrinsic and extrinsic fixed connectivity)* | | | | |
| HG | 0.00 (0.00) | 0.29 (1.00) | 0.33 (1.00) | 0.14 (1.00) |
| aST | 0.08 (1.00) | −0.29 (1.00) | 0.25 (1.00) | – |
| mST | 0.00 (0.00) | −0.40 (1.00) | −0.30 (1.00) | 0.55 (1.00) |
| pST | 0.08 (1.00) | – | 0.46 (1.00) | 0.17 (1.00) |
| *B matrix (all trials)* | | | | |
| HG | 0.00 (0.00) | – | – | – |
| *B matrix (voice trials)* | | | | |
| HG | −0.46 (0.97) | −0.89 (1.00) | – | 0.00 (0.00) |
| aST | −0.27 (0.81) | −0.43 (0.63) | – | – |
| mST | – | −0.67 (0.99) | −0.45 (0.64) | 1.21 (1.00) |
| pST | −0.78 (1.00) | – | – | −1.93 (1.00) |
| *B matrix (speech trials)* | | | | |
| HG | 0.00 (0.00) | – | 0.00 (0.00) | – |
| aST | – | 0.00 (0.00) | 0.00 (0.00) | – |
| mST | −0.52 (1.00) | – | −1.60 (1.00) | – |
| pST | – | – | 0.00 (0.00) | 0.00 (0.00) |

Expected posterior parameters (posterior probabilities in brackets) for the A matrix parameters (intrinsic and extrinsic fixed connections) and the B matrix parameters (modulation of connections); sample of $n = 52$ human participants. The matrix is organized such that the origin of connections is represented by the columns, and the target of connections is represented by the lines.
*aST* anterior superior temporal cortex, *mST* mid superior temporal cortex, *pST* posterior superior temporal cortex, *pSTS* posterior superior temporal sulcus, *HG* Heschl's gyrus.

have to note that our experiment did not demand explicit speech recognition. An explicit speech recognition task might more directly require anterior and posterior ST activity[10,17].

A further note concerns the point that we did not include the right pSTS in our DCM analysis. This was based on the fact that the right pSTS did not reach significance as peak activity location in the various GLM contrasts that we performed. This was the first reason why the right pSTS was not included in our DCM analyses, because DCM requires that neural nodes show activations (and variations of activation) according to different conditions of the experiment. Second, previous work by Venezia and colleagues[45] focused on human STS regions and they reported that specific-speech activation in the left hemisphere is more distributed and extends towards the posterior STS subregion, while the right hemisphere clustered speech-specific activity in rather anterior to mid STS subregions. Third, the pSTS is rather known as an associative processing area, and seems responsive to combined auditory and visual stimulation[46,47]. The right pSTS has been linked in particular to long-term functional reorganization upon long lasting auditory deprivation, as demonstrated in early-deaf adults[48]. Such an adaptable region might usually act as an additional hub, recruited upon other speech-related contexts. As part of a larger network involving visual information, the right pSTS might influence the micro-circuit only in the presence of additional inputs, related to speech and face recognition for instance. A recent study provides evidence in this direction as the right pSTS exhibits a strong driving role in face recognition processing[49].

Next, given this specific distribution of voice- and speech-sensitive nodes in left and right AC, we determined the local directional neural network between these nodes using DCM. Using this modeling, we could identify local neural networks that also showed some hemispheric differences. Concerning the left hemisphere, we found that aST and pSTS seemed relatively independent nodes for voice and speech processing according to the modulation of connections (Fig. 3b). We specifically found an integrated HG-ST network with two sub-branches, one for general voice processing (HG-pST) and another one for speech processing (HG-mST). The left AC thus seems to represent both independent and integrated network dynamics across low-level and high-level AC for voice and speech processing[50], which only

partly resembles commonly assumed models of processing hierarchies for sound and auditory objects in the AC[18,23]. So far, research predominantly assumes that voice processing should happen at a prior stage to speech processing, but recent studies provide contrary evidence. Previous work looked at how different subareas are responding to speech compared to other voice information[19,51], but none of these studies specifically contrasted voice (i.e., as a general auditory object) against speech processing (i.e., as a specific voice feature). Additionally, their observations rather pointed towards more complex interactions of specialized neural components, rather than a strict hierarchy of processing stages. From our observations, there seems to be partly a neural co-dependence between voice and speech processing in the left AC, such that each function can support the other by shifting important information between corresponding neural regions.

Compared to the left AC, we found a different local neural architecture and neural dynamics in the right AC with relatively disintegrated voice and speech processing pathways (Fig. 3b). The mST seemed to work as a specific node for speech processing, while aST and pST seemed to be some independent nodes for voice processing. Furthermore, voice and speech processing appeared rather separated in the right AC network in terms of modulated connections[28] (Fig. 3b), and both are primarily processed by the ST nodes as part of the higher-order AC[35]. We only found a pST-to-mST connection that seemed to be positively modulated by voice processing in general, such that speech processing in mST is potentially facilitated by voice information provided by the pST. Thus, although the right AC appears to have specific nodes for processing different kinds of voice signals, there seems to be less co-dependence of these nodes in terms of relevant information exchange in support of their functional processing. Especially, speech processing nodes seem to not share neural information with more general voice processing nodes in the right AC.

Our data overall point to some commonalities in the local AC micro-network for voice and speech processing, but they also point to considerable differences. One commonality might concern the functional contribution of the aST to the neural network architecture. The aST seems like a largely disintegrated node for voice processing bilaterally (Fig. 3b), such that voice processing in right and left AC is primarily based on an initial object classification (aST) that is then confirmed by further acoustic analysis (HG), which is rather reverse to classical hierarchical processing in AC[20]. In terms of the network differences, we found the pSTS as a relevant neural network node only in the left hemisphere. Regarding the role of the pSTS in the left hemisphere, it is likely that the modulation of connections between mST–pST–pSTS are more consistent than in the right hemisphere potentially due to this additional pSTS node. However, the differences between the two hemispheres for pST and mST are also noteworthy. The left pST showed neural integration according to a positive reciprocal modulation of connection with the HG (Fig. 3b), while the right pST was only negatively modulated by the HG. The left mST also seems more integrated in the micro-network in the left compared to the right hemisphere. The left mST was positively modulated by HG (while negative in the right hemisphere), and positively modulated the pST in turn (Fig. 3b). The left mST is differentially connected to HG in an effective manner when comparing the two hemispheres (Fig. 3a). Hence, besides the assumed influence of an additional node in the left hemisphere—contributing to a certain extent to a local cohesion—the bilateral micro-networks seem different when we consider their individual neural nodes and connections.

These data altogether point to local and rather asymmetric AC micro-networks that support both voice processing in general and speech processing in specific. There have been many hypotheses

about asymmetries of the AC in processing sound, voice, and speech. A common theory postulates a sensitivity for temporal sound information in the left AC and for spectral information in the right AC[52,53], such that the left AC might more strongly respond to speech and vocalizations, while the right AC has a higher sensitivity to frequency sweeps that might be related to prosodic elements of vocalizations[53–55]. While these observations point to functional asymmetry in AC, they do not help to predict asymmetries of neural network differences in AC[56,57]. A general observation in our data was a stronger integration of voice and speech processing in the left hemisphere, while these two types of processing were largely disintegrated in the right AC when looking at the connectivity patterns, suggesting therefore more separate pathways in the right hemisphere. This might point to a general neural network asymmetry for processing vocalizations in the primate and especially in the human brain[56].

The present findings about the left and right AC neural circuitry for voice and speech processing were based on certain methodological parameters in our study. Regarding these methodological parameters, we would like to address three potential limitations of our study First, the voice localizer task involved only the implicit processing of voice and speech sounds. During the experiment, a basic attention level was maintained by the use of a 1-back task (i.e., compare the current sound to the previous sound and press a button if a sound repeats), it was not specifically ensured that participants recognized the different sound categories correctly. However, we used a pre-evaluated set of stimuli carefully selected to achieve a high level of correct perceptual categorization[58]. Additionally, we did not aim to model some sort of specific perceptual categorization behavior during explicit sound classifications, but we were rather interested to examine which parts of the AC process voice signals in a local AC micro-network. The 1-back task was therefore only relevant to sustain attentive listening and to ensure proper neural processing of the presented sounds. This is according to earlier studies exploring voice processing by using a similar voice localizer as we used here[1,34,59]. Second, we decided to include only a limited number of connections in our DCM models instead of modeling full connectivity between all neural nodes. Connections were only allowed between neighboring nodes in each hemisphere, and no inter-hemispheric connections were modeled. While the former restriction was based on connection principles in the AC, such that AC regions predominantly communicate over short-range connections[20,23], the second restriction was based on the fact that our main focus lay on the local intra-hemispheric networks rather than inter-hemispheric connections. Third, DCM modeling is a powerful approach to model directional neural network connections, especially for event-related experimental designs[25], but it also can have some limitations when using some specific modeling approaches[60] that however can be addressed by Bayesian estimation strategies[61], especially using recent developments for PEB[33] procedure when DCM modeling is performed to comprehensively search through a large model space.

In summary, our data, first, seem to point towards bilateral local AC networks with separate but related neural pathways for the processing of different voice sounds (speech, nonverbal). A surprising finding was that neural nodes for speech processing were not necessarily located the most downstream in anterior or posterior ST but interleaved with the voice-dedicated nodes. Neural auditory stream models separating a dorsal and a ventral stream[23] usually locate speech nodes at higher levels of auditory processing remotely from primary and secondary AC regions. Here, specific speech nodes were rather surrounded by more general voice nodes, which highlights the integral co-dependency of the two micro-circuits with one another. Second, there was a hemispheric asymmetry in this local AC network. The left and

right AC showed differential nodes for voice and speech processing, and the right AC showed a higher disintegration of the voice and speech nodes, while the presumably more speech-sensitive left hemisphere showed higher neural integration for voice and speech processing. Third, voice and speech processing seems to be based on the systematic integration of low-level and high-level AC regions, predominantly in a non-hierarchical fashion. The present findings thus might challenge classical hierarchical processing models in which information is shifted from low to high-level AC regions in a serial manner, both for sound analysis and especially for auditory object recognition.

## Methods

**Participants.** The sample consisted of 52 healthy human participants (22 females, mean age 23.95 years, SD 4.22, range 18–34 years). The inclusion criteria for the experiment were normal or corrected-to-normal vision and no history of neurological or psychiatric disorders. All participants gave written informed consent and were financially reimbursed for participation. The study was approved by the cantonal ethics committee of the Swiss Cantone Geneva.

**Experimental design and task.** Stimuli were recordings of 70 voice sounds (speech, nonverbal/non-speech) and 70 non-voice sounds (animal, natural, and artificial sounds)[58]. Stimuli were presented in an event-related design, with 10% of the sounds repeated randomly for a 1-back task, where participants were asked to press a button upon a consecutive repetition of a sound. Each sound lasted 500 ms. All sounds were presented one time in a random order, with a jittered ITI of 4.0–5.5 s, and a sound intensity level of 70 dB SPL.

**fMRI data acquisition.** Functional brain data were recorded with whole-brain functional imaging data on a 3 T SIEMENS Tim Trio System (Siemens, Erlangen, Germany), using a T2*-weighted gradient multiband echo-planar imaging (M-EPI) pulse sequence (acceleration factor 4, 2 mm$^3$ isotropic resolution, 28 slices in a $64 \times 64$ matrix, 20% distance factor, TR/TE = 650/30 ms, FA 50°). We used a partial volume acquisition protocol with 28 slices, but with a higher spatial resolution. The slices were rotated ~30° to the AC–PC plane (nose-up) and covered all parts of the AC and the inferior frontal cortex. By using a higher spatial resolution (2 mm$^3$ voxels) we intended to be spatially more precise in determining and separating AC peak activations for voice and speech processing. A structural image was acquired for each participant and had 1-mm isotropic resolution (192 contiguous 1-mm slices, TR/TE/TI = 1900/2.27/900 ms, FoV 296 mm, in-plane resolution of $1 \times 1$ mm).

**Pre-processing of functional brain data.** Pre-processing of fMRI data was performed using the Statistical Parametric Mapping software (SPM12; version 7771; Welcome Trust Centre for Neuroimaging, London, UK; fil.ion.ucl.ac.uk/spm/software/spm12). Images were corrected for geometric distortions caused by susceptibility-induced field inhomogeneity[62]. A combined approach was used, which corrects for both static distortions and changes in these distortions from head motion[63,64]. The static distortions were calculated for each subject from a b0 fieldmap that was processed by the FieldMap toolbox (version 2.0, fil.ion.ucl. ac.uk/spm/toolbox/fieldmap/) as implemented in SPM12. With these parameters, functional images were then realigned and unwarped, a procedure that allows the measured static distortions to be included in the estimation of distortion changes associated with head motion. Slice time correction was performed to correct for differences in the acquisition time of individual brain slices. The motion-corrected images were then co-registered to the individuals' anatomical T1 image by using a 12-parameter affine transformation. Finally, images were deformed into the standard MNI space, using normalization parameters estimated with the CAT12 toolbox (version 12.5, neuro.uni-jena.de/cat/), and smoothed with a 6-mm FWHM isotropic Gaussian kernel to increase the signal-to-noise ratio.

**Functional TVA activation.** After pre-processing, we estimated the BOLD responses to voices and non-voices using a GLM that contained two regressors for voice sounds (speech, nonverbal) and three regressors for non-voice sounds (animal, natural, and artificial sounds), constructed by convolving a stick function at each sound onset with a canonical hemodynamic response function. Additionally, repeated sounds were modeled in a separate regressor and disregarded in further analyses. The resulting design matrix also contained a standard 128-s high-pass filter and motion estimates as covariates of no interest. Planned contrasts were then computed to create a contrast image for each of the five conditions. These contrast images were then taken to a second-level factorial group-level analysis, including the same five conditions contrasted against each other. Contrast images were thresholded at $p < 0.05$ including a voxel-wise FWE correction.

**Anatomical basis of AC connectivity for DCM.** We built the DCM models by combining anatomical and functional information about the AC connectivity as

reported previously. For better comparability, the terms used to describe regions in the respective papers are related to the anatomical terms used here in our study. From a neuroanatomical perspective and given its primary role in auditory processing, HG specifically represents the intersection area of white fibers integrating the auditory areas to neighboring regions. One of the pathways for auditory integration involves U-fibers connecting the HG to the middle temporal gyrus (MTG), but a bundle of U-fibers also connects the superior temporal gyrus (STG) with the MTG[30]. Detailed connectivity investigations through in vivo tractography of the low-order auditory areas unravel a hierarchical organization in three stages, comparable to nonhuman primates' structure[65], with a core (including the primary auditory cortex (PAC)—here referred to as HG), belt (areas immediately adjacent to the HG), and parabelt regions (i.e., anterior parabelt—here referred to as aST, superior temporal area (STA)—here referred to as mST, and posterior parabelt—here referred to as pST). Indications for a resemblance of the AC organization across the Old World monkey species are constantly growing: mapping the human auditory areas based on cytoarchitectonic features for instance largely support the parallel made between macaques and humans for the conserved existence of the different subparts of the auditory core, belt and parabelt regions[35,66].

Generally, the structural connections on the medial–lateral axis would be core-to-belt and belt-to-parabelt directions (both reciprocally). From this perspective, structural connections exist between HG and our ST regions through an intermediate stage, which guides the flow of auditory information in a cascade-like fashion[29]. These observations largely coincide with previous work by Upadhyay and colleagues who identified effective connectivity between the PAC (caudal and rostral HG—here referred to as HG) and PSTG (pST and lateral planum temporale —here referred to as pST) and between the PAC and ASTG (aST and lateral planum polare—here referred to as aST) suggesting that those regions are parts of the same auditory processing circuitry[28]. Yet, instead of the three-tier architecture of the auditory cortical area described above and shown in the nonhuman primate organization of auditory processing, only two of the three stages were detected in our study in terms of functional activations. Earlier studies reported similar observations, strongly arguing that regions tagged as the belt area might not be sensitive to passive voice listening tasks[28].

Furthermore, evidence for structural connections between the three voice-sensitive patches across the STS (posterior, middle and anterior right STS) in humans have already been unraveled in vivo through tractography[47]. Such methodology also allowed more detailed neuroanatomical investigations revealing the existence of structural connectivity as a bridge between annectant gyri within the complex topology of the STS (superior and middle temporal gyri), a compelling illustration of the high density of local connectivity, particularly through U-shaped fibers[67].

Hence, the DCM models included here (see below) were constructed in consistency with this previous work mentioning the existence of anatomical projections within the early stage of the human auditory area but also their functional expression within an integrative auditory neuronal connectome.

**Definition of VOIs and time-series extraction.** Based on the group-level brain activation peaks found in the three specific contrasts, we defined five peak locations in the left hemisphere (HG, aST, mST, pST, pSTS) and four peak locations in the right hemisphere (HG, aST, mST, pST) (Table 1). The location of each peak location within the AC areas was determined according to the Anatomy toolbox (version 2.2b, fz-juelich.de/SharedDocs/Downloads/INM/INM-1/DE/Toolbox/Toolbox_22.html) as implemented in SPM12. Peaks inside areas Te1.0–1.2 were defined as HG[38,68] and for the peaks on the ST or in the STS it was determined if they lay within the higher-order area Te3[35]. Around each location, voxels within a sphere of a 3 mm radius were selected. Only voxels that showed significant activation ($p < 0.05$, uncorrected) by the presentation of *all sounds* within each participant were included, and the voxel selection thus was adapted to individual activation profiles in each participant. This selection process was based on a separate GLM with three regressors: One regressor defined by *all sounds* modeling the onset of each sound (including voice and non-voice sounds), one regressor defined by *voice sounds* including voice sounds only (speech, nonverbal), and one regressor specifying *speech sounds*. The first regressor was chosen because it represents the overall stimulation with sounds in the experiment (i.e., all sound stimulations) which is often included as a general condition that defines overall brain stimulation and input during the experiment[25]. The other two regressors were included because they figured as the two major conditions that revealed specific activity in left and right AC during the calculation of the original contrasts (Fig. 1). This GLM contained the same regressors-of-no-interest as the previous GLM to locate voice-sensitive voxels.

From each volume of interest (VOI), the first principal component was extracted for the DCM analysis and adjusted for the F-contrast modeling the *all sound*, *all voice*, and *speech sounds* regressors. Time series data from VOIs associated with the above regressors were summarized using the SPM12 eigenvariate toolbox. Peak locations for VOI definition were ensured to be >12 mm apart from each other based on a Euclidian distance measure to ensure non-overlapping voxels in the VOIs and to take into account the smoothing kernel (6 mm) for functional brain data. We used the [voice>non-voice] contrast to define VOIs that represent voice processing (bilateral aST and pST), the [speech>nonverbal] contrast defined VOIs for speech processing (left mST and

pSTS, right mST), and a T-contrast across all five conditions (i.e., higher activity in all conditions compared to baseline) defined areas in the HG as a general sound processing VOI in bilateral low-level AC.

**DCM for effective functional connectivity.** Based on the time-series extraction in the VOIs, we subsequently applied DCM[25,69] to model effective functional connectivity between these VOIs. DCM tries to explain the observed brain responses in terms of underlying causal interactions between different areas at the neuronal level. DCM estimates the experimental modulation of (intrinsic) self-connections or (extrinsic) forward and backward connections between VOIs that are active during voice and speech processing in a directional manner. We created and estimated DCMs with the DCM12 toolbox (version 7479) as implemented in SPM12. The DCMs were based on five spherical VOIs in the left hemisphere and four VOIs in the right hemisphere, each centered on a peak located in the TVA and low-level AC (Table 1), with a radius of 3 mm to avoid possible overlap between VOIs. Separated models were generated in the right versus left hemisphere as the VOIs that were entered into the models were based on non-symmetrical peaks of activations across hemispheres. We therefore refrained from artificially creating symmetrical models for left and right auditory areas by selecting only symmetrical activation peaks in the left and right AC, as this would serve against our aim to test the functional hierarchy within the local micro-network of each hemisphere independently.

For each participant and brain hemisphere, we first created a full connectivity model (full model) with bidirectional connections between neighboring VOIs in each hemisphere (A matrix) (Fig. 2). Driving input (C matrix) to each node was specified by the experimental condition that elicited the original activity: The *all sounds* regressor provided input to HG, the *voice sound* regressor provided input to aST and pST, while the *speech sound* regressor provided input to mST (and additionally to left pSTS). The driving inputs were mean-centered, causing the parameters of the A matrix to represent the mean connection strengths across conditions. Finally, the modulation of intrinsic and extrinsic connections by experimental conditions (B matrix) followed the activation profile of the VOIs: Intrinsic connections in nodes were modulated by the activation profile of the node (e.g., intrinsic HG connectivity is modulated by *all sounds* trials), connections from ST/STS regions to and from HG were set to be modulated by the ST/STS activation profile (e.g., HG–aST connections could only be modulated by the *voice sound* trials), and forward connections from ST/STS regions to other ST/STS region were set to be modulated by the activation profile of the origin of the connections (e.g., connections originating from mST could only be modulated by *speech sound* trials). We estimated these full DCM models for each participant using Bayesian model inversion.

To estimate group-level parameters in these left and right AC networks, we conducted a second-level PEB analysis, separately for the left and right AC networks. The PEB analysis included a hierarchical model of the connectivity parameters, including connectivity parameters from all participants at the first-level (i.e., DCMs are fitted to each participants data, and posterior probability density over the parameters and the free energy taken to the group level) as well as a GLM modeling at the second-level (i.e., with one regressor per covariate per connection), and the estimation procedure used a variational scheme. After estimating the parameters of the full PEB model, we subsequently pruned away parameters using a BMR approach performed on the A matrix and B matrix parameters. The BMR performs an automatic (greedy) search over the model space to optimize model evidence. We here used the BMR as an exploratory approach with only minimal constraints and performed an automatic search over reduced PEB models. This search was accomplished with the simplifying assumption that these models were all equally likely a priori. The model evidence takes into account both model accuracy (how well the model fits the data) and model complexity (the difference between model parameters and their prior values). The BMR procedure was accomplished in an iterative process. Finally, we applied BMA across all models searched by the BMR, and averaging was performed as a weighted average of parameters across models according to the posterior probabilities of the models. Significant network parameters were determined with a posterior probability of $p > 0.99$.

**Statistics and reproducibility.** Detailed information on statistical tests is provided in the respective subsections. We analyzed fMRI data with the Statistical Parametric Mapping software (SPM12; version 7771) according to current standard procedure in fMRI data analysis by using a GLM, with separate regressors for each sound category, based on stick functions at each sound onset convolved with a canonical HRF. We report functional activations with a voxel threshold of $p = 0.05$, corrected for multiple testing by controlling the family-wise error rate (FWE) to avoid false-positive activations. The FWE method is especially appropriate and efficient to control for potential false positive activations in functional neuroimaging data[70].

The functional connectivity analysis was performed in a Bayes statistical framework using the PEB approach and using standard priors as implemented in the DCM12 toolbox package (version 7479) in SPM12. We used the functional activations and previous evidence about anatomical connections to build probable connectivity models separately for each hemisphere. However, especially the BMR approach was setup as an exploratory approach with the

simplifying assumption that all potential models were all equally likely a priori. We chose to use an exploratory approach, since the exact prediction about each connectivity parameter was rather difficult to establish given previous evidence, and thus an a priori choice of models would have been potentially biased. The significance of the resulting network parameters was determined with a posterior probability of $p > 0.99$.

In accordance with the APA style, we report $t$-values with degrees of freedom for reporting the functional activations when contrasting experimental conditions. For the PEB analysis, we report expected posterior parameters and their posterior probabilities resulting from the Bayes statistical approach. Sample sizes are reported in detail in each figure legend, the main text, and the corresponding method sessions.

We expect that all data should be reproducible if followed with the same settings and procedures as described in the "Methods" section and reporting summary and using the same stimulus material.

**Reporting summary**. Further information on research design is available in the Nature Research Reporting Summary linked to this article.

## Data availability

The unthresholded SPMs of all contrasts displayed in Fig. 1 were uploaded to NeuroVault: https://identifiers.org/neurovault.collection:9707. The DCM data are available on OSF: https://osf.io/8t3aw/. The ethical approval for this study and legal restrictions in Switzerland do not allow us to share raw data openly. The raw data that support the findings of this study can be made available from the corresponding author upon a reasonable request and in consultation with the ethical committee of the Canton Geneva (Switzerland). The sound material used in this study for auditory stimulation was provided by Capilla and colleagues[58].

## Code availability

The DCM code is available on OSF: https://osf.io/8t3aw/. No other custom code or mathematical algorithms were used in the study. All software used for statistical analyses has been declared in the manuscript.

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

## Acknowledgements

The study was supported by the Swiss National Science Foundation (SNSF PP00P1_157409/1 and PP00P1_183711/1 to S.F.). S.F./M.B. also received grant support from the Vontobel foundation (www.vontobel-stiftung.ch; Zurich, Switzerland).

## Author contributions

F.S. was involved in data analysis, and writing of the manuscript; M.B. was involved in data analysis, and writing of the manuscript; S.F. was involved in study design, data acquisitions, data analysis, and writing of the manuscript.

## Competing interests
The authors declare no competing interests.

## Additional information

Primary Handling Editors: Jeanette Mumford and George Inglis

