## [Peer Review File · Communications Biology]

Reviewers' comments:

Reviewer #1 (Remarks to the Author):

This study investigated heterogeneity within the voice area of the auditory cortex. The authors applied connectivity analysis (dynamic causal modelling, DCM) to fMRI data that they acquired in a sample of healthy subjects. They found that different subregions of auditory cortex, and particular connections among those subregions, responded distinctly to speech and non- speech vocal sounds. The study was conducted to a high standard, with best practice followed for the fMRI analysis and subsequent connectivity analysis. My main suggestion for improving the study is to set out some specific hypotheses regarding network architectures, and explicitly test those hypotheses using DCM. At present, an automatic search over candidate models was conducted, and post-hoc interpretation provided. This means there is not currently a strong connection between the author's hypotheses (which relate to hierarchical vs non hierarchical structure of the auditory cortex) and the DCM analysis that was performed.

In more detail, in the authors' models, the effect of the two experimental conditions could enter the network directly via any of the five regions, as well as being able to modulate the strength of connections between those regions. That required a large number of parameters, encompassing thousands of possible hypotheses about how the network might be organised (here I'm considering one "hypothesis" to be the presence vs absence of an experimental condition on a particular connection). This is not wrong, but it is unfocussed, and assumes that all potential reduced models were equally likely a priori. A more robust approach, which is likely also to be more replicable, would be to compare the evidence for some specific mixtures of these parameters - i.e. to compare specific PEB models embodying different architectures. Ideally, the introduction would present a small number of hypothesised networks, and the results would provide a Bayesian model comparison showing the posterior probability for each network, providing a clear narrative and take-home message for the paper. This would not require re-specifying the DCMs - only a revised second level analysis.

One further minor note for the modelling aspect of the paper. It seems quite ambitious to have estimated the effect of each experimental condition as both modulating connections (B-matrix parameters) as well as driving regions aST, mST, pST and pSTS directly (C-matrix parameters). I imagine that this would have introduced some quite profound covariance among the DCM parameters, which could have reduced the efficiency of model inversion. If the authors are happy with their results, then this has not been a problem. Otherwise, it would be worth having speech and voice only as modulatory inputs in the DCMs, but not as driving inputs on aST, mST, pST and pSTS, in order to simplify the model.

Minor points

Page 2 line 30 - extension -> extent

Page 2 line 34 - neuroimaging -> functional MRI (fMRI)

Page 2 line 35 - in specific -> in particular (or specifically)

Page 5 line 79 - descoing, was decoding meant here?

Page 15 line 365 - to be pedantic, these are "simple mains effects" or "simple effects" rather than main effects (the main effect is voice vs non-voice)

Reviewer #2 (Remarks to the Author):

The article reports on a Dynamic Causal Modelling (DCM) for regions within Temporal Voice Areas (TVA). I found the article methodologically sound as well as well written and clear (given the complexity of DCM). I have however a few suggestions – which I see as major (although minor editing wise).

1 – There is no Voice Area (VA) – even the authors make the point of bilateral activations, which would be voice areas. In addition, since the analysis is limited to the temporal cortex, I strongly recommend using the terminology proposed by P. Belin, as reported in Pernet et al., i.e. Temporal Voice Areas (TVA).

2 – Results and figure 1. (A) An optional quick analysis to do, is to report how much overlap you have in your fig.1 with the 'canonical' map <https://neurovault.org/images/106/> - which tells the reader that we are for sure in the same regions. (B) The figure 1 needs rain plots for each ROI, showing activations levels of each regressor. (C) the statistical (unthresholded) map needs to be uploaded on NeuroVault.

3 - DCM and ROI: How was Heschl Gyrus (HG) ROI defined? there are well reported regions in the article, also explaining TE1,2,3 which I'm guessing were visualized / checked using the anatomy toolbox – if that is the case mention it in the method and reference the tool and maps, if not how did you ensure you are in the regions described.

4 – introduction and discussion – the whole point of the paper is looking at processing of speech vs. voice, yet there is little reference to this. For instance, in <https://pubmed.ncbi.nlm.nih.gov/26247409/> we clearly dissociated the two, with also a difference left/right hemispheres – although in patients this related to right frontal damages which we proposed reflected a dissociation to right (anterior?) TVA ; which seems to fit with what is observed here. Similarly, the co-optation hypothesis seems to explain left TVA (the more general question under the co-optation hyp. is the why an asymmetry). I'd suggest having a broader view on voice per se vs. other features (being speech, but also emotion, gender, identify).

5 – material availability: define 'reasonable request' – IMO this is not good enough. I do understand Swiss law makes it impossible to share openly raw MRI images. It should be explained as such, and under which conditions this could be shared if someone ask. Indicate whom would be in charge of that. The derived data must be shared – SPM of fig.1 on neurovault and ROI with DCM modelling/code in a repository. Those are clearly not identifiable, and there are thus no legal issues. If not shared either, justify it.

Reviewer #3 (Remarks to the Author):

Reviewing of „Auditory cortical micro-networks for voice processing“ by Steiner et al. In an fMRI study the authors investigate local microcircuits of the voice area involved in processing voice of speech and non-speech stimuli and their interactions. The goal of the study is to further elucidate the homogeneity/heterogeneity of the voice processing area. They first identify nodes within the voice area by contrasting voice/non-voice stimuli (left and right aST, pST) and speech/non-voice stimuli (left mST, pSTS and right mST). Second, they perform a DCM and PEB model reduction analysis based on these nodes to reveal the network interactions separately for the left and right hemisphere in terms of effective connectivity and connection modulation. They find different connectivity patterns in the left and right hemisphere, and areas where only modulation effects from higher to lower level areas were observed (aST to HG). The authors interpretation is that voice and speech processing recruit bilateral non-hierarchical networks in AC. Voice and speech processing networks are more integrated in the left AC, and rather disintegrated in the right AC.

The interaction and distinction of voice and speech-specific areas is exciting, as it might elucidate the mechanisms for speech intonation vs. voice identity processing. Advancing the knowledge of the micro-network architecture of voice processing is relevant, and a basis for a better understanding of its role in speech. This is a very well written and interesting manuscript. The hypotheses are clear. The analysis are advanced and limitations are discussed. I have some rather minor comments.

Detailed comments

- To what extent would the authors assume do the connectivity results change if a symmetrical model is used (adding the pSTS to the right hemisphere?)
- L. 238: "especially if one assumes that voice processing needs to happen at a prior stage to speech processing"; As I understand theories such as the AST (Poeppel, 2003), intonation and acoustic phoneme processing can occur in parallel. A possible limitation here seems that the contrast speech vs. no-voice was used as basis for the modeling. It is possible that to some extent speech-specific processing that does not reflect voice aspects is reflected in the results; (the authors also show the contrast speech vs. non-speech voice, with similar albeit less strong activations)
- l. 650: I assume the contrast is speech vs. non-verbal but voice material, could you clarify in the legend?
- Discussion: l. 245-248: this seems to refer to the modulations Fig. 3 B, which seem similar for aST in the left and the right. Why is the disintegration for the right aST emphasized, while some other modulations seem to change more? The stronger interaction between voice and speech areas in the left hemisphere seems particularly due to the additional speech node pSTS?
- In the discussion: the interpretation of the effective connectivity and modulation of connection results could at times be more clearly distinguished (i.e. what part of the discussion is referring to which of the findings)
- Figure 3: flipping the right or left hemisphere, might make it easier to compare the network structures?

Wording

- L. 56-60: the sentence structure is quite complicated, could you simplify for readability? Same in line 66-69
- L. 88 "sound"

Reviewer #1 (Remarks to the Author):

This study investigated heterogeneity within the voice area of the auditory cortex. The authors applied connectivity analysis (dynamic causal modelling, DCM) to fMRI data that they acquired in a sample of healthy subjects. They found that different subregions of auditory cortex, and particular connections among those subregions, responded distinctly to speech and non-speech vocal sounds. The study was conducted to a high standard, with best practice followed for the fMRI analysis and subsequent connectivity analysis.

Response: We thank reviewer #1 for this overall very positive evaluation of our manuscript and the suggested ideas about the DCM data analysis.

My main suggestion for improving the study is to set out some specific hypotheses regarding network architectures, and explicitly test those hypotheses using DCM. At present, an automatic search over candidate models was conducted, and post-hoc interpretation provided. This means there is not currently a strong connection between the author's hypotheses (which relate to hierarchical vs nonhierarchical structure of the auditory cortex) and the DCM analysis that was performed.

Response: We very much thank the reviewer for suggesting this alternative approach for running DCM analyses on neural brain data. DCM analyses typically can be performed in two major ways; (1) With a limited number of brain nodes included, some specific a priori hypotheses can be formulated and tested with a limited but well-informed number of DCM models (usually in the range <20 models). Based in prior evidence, some specific DCM models are formulated. This approach has the downside that only models will be tested for which some prior evidence exists. (2) In an unbiased approach, one can use a Bayes estimation approach to exhaustively search a large model space that considers any possible network architecture. The winning model is the one that fits the data the best. This approach has the advantage of not being biased by prior knowledge and a researcher-bias for an *a priori* selection and limitation of the model space. One downside might be that very specific hypothesis could not be formulated in this case given the number of parameters.

In the case of our study, we chose the second option, because we thought that an unbiased Bayesian search over the model space would reveal more interesting and maybe "new" results than staying within an a priori and existing model space that is limited by prior studies. We have to note however, that even this unbiased procedure was not accomplished without evidence-based constraints in the neural network modelling. First, our DCM modelling approach included only neural connections between nodes for which anatomical evidence was existing. Second, driving inputs and modulation of connections was only allowed for conditions, which elicited significant activity in the chosen neural nodes.

Overall, we think this is a valid DCM modelling approach, which has been similarly used in previous studies on the auditory cortex ¹. We had two major and differential research questions:

(1) integrated vs. disintegrated processing of voice and speech sounds in the auditory cortical micro-network, (2) hierarchical vs non-hierarchical processing of voice sounds in general and speech sounds in specific.

These two major questions are described in p5:

“We tested two main questions in our study: our first question concerned the notion of rather integrated or rather separated neural pathways for voice signal processing and voice signal differentiation in the AC. By integrated neural pathways, we refer to the potential observation that different types of voice signals (voice signals in general, speech signals) are processed along similar neural pathways in the AC, and also that this processing integrates the distributed information decoding across different local AC processing nodes. Contrarily, separated neural pathways would show disintegrated processing of various voice signals, separated across different neural AC nodes. Our second question concerned the notion if cortical voice processing follows documented functional hierarchies in the AC for auditory object processing ². A typical processing hierarchy in the AC includes an information flow from primary, secondary, to higher-level AC regions, which has been largely demonstrated for sound feature processing and the processing of simple auditory objects ². The neural treatment of rather complex and socially relevant voice sounds might however also include a non-hierarchical AC processing, which dynamically shifts relevant information between low- and higher-order AC regions ^{3,4} for sounds analysis as well as voice detection and voice type discrimination, respectively.”

As we describe below, we think that our approach is most sensitive to the question we asked in the paper, and it provides the required details to model the complex connectivity in the auditory cortex for voice and speech object processing. Especially the analysis approach we pursued is most appropriate to determine auditory cortical micro-networks. Integrating our two major research questions into an approach with pre-selected models would also result in a large number of hypothesized networks beyond a level that could be easily demonstrated in the paper.

In more detail, in the authors' models, the effect of the two experimental conditions could enter the network directly via any of the five regions, as well as being able to modulate the strength of connections between those regions. That required a large number of parameters, encompassing thousands of possible hypotheses about how the network might be organised (here I'm considering one "hypothesis" to be the presence vs absence of an experimental condition on a particular connection). This is not wrong, but it is unfocussed, and assumes that all potential reduced models were equally likely a priori. A more robust approach, which is likely also to be more replicable, would be to compare the evidence for some specific mixtures of these parameters - i.e. to compare specific PEB models embodying different architectures. Ideally, the introduction would present a small number of hypothesised networks, and the results would provide a Bayesian model comparison showing the posterior probability for each network, providing a clear narrative and take-home message for the paper. This would not require re-specifying the DCMs - only a revised second level analysis.

Response: Following up to our response to the previous comment of the reviewer, we feel that we need to clarify some important features of DCM modelling in a PEB framework. There are five important features.

First, DCM-PEB modelling is as much as replicable as any other brain data analysis. It might be even more robust given that DCM-PEB works simultaneously on the single-subject and group-level, switching the Bayes estimation process repeatedly between both levels. Given this, each parameter thus is also not "... one separate hypothesis ...", but only meaningful when estimated with the other parameters.

Second, and relatedly, DCM-PEB is not a two-step procedure (single-subject modelling and estimation, followed by group-level estimations), but simultaneously models and estimates parameters on the subject/group level. So, there is no separate second-level analysis.

Third, the PEB approach is used to compare models, but is not used as a DCM model itself. So, one cannot "... compare specific PEB models ...", but PEB is a procedure used in Bayesian model reduction (BMR).

Fourth, although DCM-PEB assumes that all reduced models are equally likely *a priori*, this changes over the course of the estimation procedure. DCM-PEB implements a Bayesian model reduction (BMR) approach where recursive estimations on and elimination of parameters become part of the estimation procedure as priors. The final model is not a "random" model out of equally likely *a priori* models, but the model that fits the data the best after many recursive estimation processes.

Fifth, the final model that fits the data the best and that we report in the paper is not single model out of "... thousands of possible hypotheses ..." but is the weighted average of all models included in the PEB approach. As we described in the methods section, the final step of PEB is using a Bayesian Model Averaging (BMA) approach, which creates the average of all models weighted by their posterior evidence. We think that this procedure is a much more sophisticated approach than choosing a limited number of models *a priori*, and then only reporting the winning model. We also feel that this approach reveals much more replicable and robust data than any selective modelling approach.

Again, we have to note that our DCM modeling approach was not unconstrained in a sense that all variations on each parameter was allowed. We introduced constraints in the possible connections based on prior anatomical evidence, and connections could be only modulated by a condition that elicited activity in the source node in our GLM based data analysis.

P6: "We tested left and right AC functional connectivity models on a local micro-network level with the following constraints: (a) We allowed bidirectional connections between neighboring nodes in the AC, given that AC regions predominantly communicate with neural nodes in short-range connections^{2,5}; (b) driving inputs to neural nodes were defined by dominant experimental conditions into each node; and (c) modulation of connections between nodes by specific voice and speech sound condition".

We also included a more detailed description of the DCM-PEB approach on p6: "The PEB procedure includes three important subcomponents in a Bayes framework to search the connectivity model space and to estimate network parameters: First, it simultaneously models and estimates parameters on the participant and group data level; second, using a Bayesian Model Reduction (BMR) approach the PEB approach prunes away rather unimportant and non-consistent network parameters as quantified by their posterior probability; and third, using a Bayesian Model Averaging (BMA) approach, the final model and its network parameters (i.e. posterior probabilities) result from a the weighted averaging of these parameters, and thus takes

into account all estimated models from the BMR approach leading to rather robust representation of neural network properties. Based on this DCM-PEB modelling approach, the resulting connectivity patterns were expected to represent robust functional pathways for voice and speech processing in local AC networks in the left and the right hemisphere”.

P6: “The PEB procedure includes three important subcomponents in a Bayes framework to search the connectivity model space and to estimate network parameters: First, it simultaneously models and estimates parameters on the participant and group data level; second, using a Bayesian Model Reduction (BMR) approach the PEB approach prunes away rather unimportant and non-consistent network parameters as quantified by their posterior probability; and third, using a Bayesian Model Averaging (BMA) approach, the final model and its network parameters (i.e. posterior probabilities) result from a the weighted averaging of these parameters, and thus takes into account all estimated models from the BMR approach leading to rather robust representation of neural network properties. Based on this DCM-PEB modelling approach, the resulting connectivity patterns were expected to represent robust functional pathways for voice and speech processing in local AC networks in the left and the right hemisphere.”

One further minor note for the modelling aspect of the paper. It seems quite ambitious to have estimated the effect of each experimental condition as both modulating connections (B-matrix parameters) as well as driving regions aST, mST, pST and pSTS directly (C-matrix parameters). I imagine that this would have introduced some quite profound covariance among the DCM parameters, which could have reduced the efficiency of model inversion. If the authors are happy with their results, then this has not been a problem. Otherwise, it would be worth having speech and voice only as modulatory inputs in the DCMs, but not as driving inputs on aST, mST, pST and pSTS, in order to simplify the model.

Response: The reviewer mentions an important point here, but covariance has not been reported to be a major issue within DCM modelling, especially in a Bayes framework. Also, DCM modelling including only a B-matrix without a C-matrix is in principle not possible; the network needs driving input to set the network in an “active” state.

Minor points

Page 2 line 30 - extension -> extent

Response: Was changed accordingly.

Page 2 line 34 - neuroimaging -> functional MRI (fMRI)

Response: Was changed accordingly.

Page 2 line 35 - in specific -> in particular (or specifically)

Response: Was changed accordingly.

Page 5 line 79 - descoing, was decoding meant here?

Response: Yes, “decoding” was meant and changed accordingly.

Page 15 line 365 - to be pedantic, these are “simple mains effects” or “simple effects” rather than main effects (the main effect is voice vs non-voice)

Response: The sentence was changed as follows, p17: “Planned contrasts were then computed to create a contrast image for each of the five conditions.”

Reviewer #2 (Remarks to the Author):

The article reports on a Dynamic Causal Modelling (DCM) for regions within Temporal Voice Areas (TVA). I found the article methodologically sound as well as well written and clear (given the complexity of DCM). I have however a few suggestions – which I see as major (although minor editing wise).

Response: We thank reviewer #2 for this overall very positive evaluation of our manuscript and the positive and constructive comments.

1 – There is no Voice Area (VA) – even the authors make the point of bilateral activations, which would be voice areas. In addition, since the analysis is limited to the temporal cortex, I strongly recommend using the terminology proposed by P. Belin, as reported in Pernet et al., i.e. Temporal Voice Areas (TVA).

Response: The reviewer is correct that there might be not a single TVA, but maybe a collection of many TVAs. This is of course the terminology introduced by Belin/Pernet in 2015⁶, showing that the auditory cortex in each hemisphere might host up to three separate voice patches. A major shortcoming of this previous approach might have been that, although it showed separate TVAs, it was very vague on the functional distinction of these different TVAs beyond a general sensitivity to voice compared to non-voice sounds.

Given the assumption that indeed different TVAs might exist, we changed our terminology throughout the manuscript to largely talk about TVAs instead of talking only of one TVA. We also changed the terminology from “voice area VA” to “temporal voice area(s) TVA”.

2 – Results and figure 1. (A) An optional quick analysis to do, is to report how much overlap you have in your fig.1 with the ‘canonical’ map <https://neurovault.org/images/106/> - which tells the reader that we are for sure in the same regions. (B) The figure 1 needs rain plots for each ROI, showing activations levels of each regressor. (C) the statistical (unthresholded) map needs to be uploaded on NeuroVault.

Response: (A) This is an interesting proposition and we accordingly compared our activations with the TVA probability map of Pernet and colleagues ⁶. All coordinates reported in our paper were overlapping with the TVA probability map and we added a sentence clarifying this to the manuscript:

P10: “These activation patterns are similar to previous reports ^{3,6}, such that all peak coordinates were completely overlapping with a previously defined TVA probability map ⁶ (<https://neurovault.org/images/106/>)”.

(B) Because there are many regions and five regressors per region, we opted for bar plots, showing the beta estimates of all conditions in each ROI, thus avoiding an overly cluttered figure. They were added as an additional panel to Fig. 1. The panel was described in the legend and referenced in the text:

P7: “For each of those peak locations we created a ROI (a sphere of 3 mm radius around the peak) to be used in the DCM analysis. Fig. 1d shows the beta estimates (i.e. the level of activation) in each condition for each of the ROIs”.

(C) The unthresholded SPMs of all contrasts displayed in Fig. 1 were uploaded to NeuroVault: <https://identifiers.org/neurovault.collection:9707>

3 – DCM and ROI: How was Heschl Gyrus (HG) ROI defined? there are well reported regions in the article, also explaining TE1,2,3 which I’m guessing were visualized / checked using the anatomy toolbox – if that is the case mention it in the method and reference the tool and maps, if not how did you ensure you are in the regions described.

Response: As the reviewer mentions correctly, all regions in the paper were checked using the Anatomy toolbox. We agree that this should be mentioned and, thus, included this information in the methods part of the manuscript (p19):

“The location of each peak location within the auditory cortex areas was determined according to the Anatomy toolbox (version 2.2b) as implemented in SPM12. Peaks inside areas Te1.0-1.2 were defined as HG ^{7,8} and for the peaks on the ST or in the STS it was determined if they lay within the higher-order area Te3 ⁹”.

4 – introduction and discussion – the whole point of the paper is looking at processing of speech vs. voice, yet there is little reference to this. For instance, in <https://pubmed.ncbi.nlm.nih.gov/26247409/> we clearly dissociated the two, with also a difference left/right hemispheres – although in patients this related to right frontal damages which we proposed reflected a dissociation to right (anterior?) TVA; which seems to fit with what is observed here. Similarly, the co-optation hypothesis seems to explain left TVA (the more general question under the co-optation hyp. is the why an asymmetry). I’d suggest having a broader view on voice per se vs. other features (being speech, but also emotion, gender, identify).

Response: We thank the reviewer for this suggestion. There is indeed previous work investigating the brain activity for speech vs. other voice components which deserve to be mentioned in our discussion. However, with the present study, we aim to add a genuine comparison of speech vs. voice and voice vs. other sounds at the scale of micro-networks in the AC. This precision has

been added, alongside the points mentioned in the above comment, in an adapted version of our discussion (p12):

“So far, research predominantly assumes that voice processing should happen at a prior stage to speech processing, but recent studies provide contrary evidence. Previous work looked at how different subareas are responding to speech compared to other voice information^{10,11}, but none of these studies specifically contrasted voice (i.e. as a general auditory object) against speech processing (i.e. as a specific voice feature). Additionally, their observations rather pointed towards more complex interactions of specialized neural components, rather than a strict hierarchy of processing stages. From our observations, there seems to be partly a neural co-dependence between voice and speech processing in the left AC, such that each function can support the other by shifting important information between corresponding neural regions.”

5 – material availability: define ‘reasonable request’ – IMO this is not good enough. I do understand Swiss law makes it impossible to share openly raw MRI images. It should be explained as such, and under which conditions this could be shared if someone ask. Indicate whom would be in charge of that. The derived data must be shared – SPM of fig.1 on neurovault and ROI with DCM modelling/code in a repository. Those are clearly not identifiable, and there are thus no legal issues. If not shared either, justify it.

Response: The unthresholded SPMs of all contrasts displayed in Fig. 1 were uploaded to NeuroVault: <https://identifiers.org/neurovault.collection:9707>

The DCM data and the DCM code were put on OSF: <https://osf.io/8t3aw/>. For the DCM modeling we followed the general and simply to follow guidelines as described here: [https://en.wikibooks.org/wiki/SPM/Parametric_Empirical_Bayes_\(PEB\)](https://en.wikibooks.org/wiki/SPM/Parametric_Empirical_Bayes_(PEB))

Reviewer #3 (Remarks to the Author):

Reviewing of „Auditory cortical micro-networks for voice processing” by Steiner et al. In an fMRI study the authors investigate local microcircuits of the voice area involved in processing voice of speech and non-speech stimuli and their interactions. The goal of the study is to further elucidate the homogeneity/heterogeneity of the voice processing area. They first identify nodes within the voice area by contrasting voice/non-voice stimuli (left and right aST, pST) and speech/non-voice stimuli (left mST, pSTS and right mST). Second, they perform a DCM and PEB model reduction analysis based on these nodes to reveal the network interactions separately for the left and right hemisphere in terms of effective connectivity and connection modulation. They find different connectivity patterns in the left and right hemisphere, and areas where only modulation effects from higher to lower level areas were observed (aST to HG). The authors interpretation is that voice and speech processing recruit bilateral non-hierarchical networks in AC. Voice and speech processing networks are more integrated in the left AC, and rather disintegrated in the right AC.

The interaction and distinction of voice and speech-specific areas is exciting, as it might elucidate the mechanisms for speech intonation vs. voice identity processing. Advancing the knowledge of

the micro-network architecture of voice processing is relevant, and a basis for a better understanding of its role in speech. This is a very well written and interesting manuscript. The hypotheses are clear. The analyses are advanced, and limitations are discussed. I have some rather minor comments.

Response: We thank reviewer #3 for this overall very positive evaluation of our manuscript.

Detailed comments

- To what extent would the authors assume do the connectivity results change if a symmetrical model is used (adding the pSTS to the right hemisphere?)

Response: We thank the reviewer for this interesting query. Actually, this question directly reflects on the fMRI results in the first part of the study, showing an asymmetry in the regions later included for the DCM model in the left vs. right hemisphere. The functional peak activations revealed by the GLM contrasts [voice > non-voice] and [speech > nonverbal] constituted our basis for modeling the micro-circuits, and while the left pSTS is present, the activity within the right pSTS is not sufficient to survive the family error-wise correction applied to test the strength of BOLD activation. This was the first reason why the right pSTS was not included in our DCM analyses, because DCM requires that neural nodes show activations (and variations of activation) according to different conditions of the experiment.

It is delicate to speculate on the importance that the right pSTS would have had if included in the local micro-circuit. Possibly, the right pSTS is not specific enough to be included at the same level as other subregions such as the mST. We give detailed argumentations now in the manuscript

P11: "A further note concerns the point that we did not include the right pSTS in our DCM analysis. This was based on the fact that the right pSTS did not show up as peak activity location in the various GLM contrasts that we performed. This was the first reason why the right pSTS was not included in our DCM analyses, because DCM requires that neural nodes show activations (and variations of activation) according to different conditions of the experiment. Second, previous work by Venezia and colleagues⁴² focused on human STS regions and they reported that specific-speech activation in the left hemisphere is more distributed and extends towards the posterior STS subregion, while the right hemisphere clustered speech-specific activity in rather anterior to mid STS subregions. Third, the pSTS is rather known as an associative processing area, and seems responsive to combined auditory and visual stimulation^{43,44}. The right pSTS in particular has been linked to long-term functional reorganization upon long lasting auditory deprivation, as demonstrated in early-deaf adults⁴⁵. Such an adaptable region might usually act as an additional hub, recruited upon other speech-related contexts. As part of a larger network involving visual information, the right pSTS might influence the micro-circuit only in the presence of additional inputs, related to speech and face recognition for instance. A recent study provides evidence in this direction as the right pSTS exhibits a strong driving role in face recognition processing⁴⁶".

For the left hemisphere, our data depict the pSTS as a speech node positively influenced by the neighboring region, namely the pST (a voice node), and negatively influencing both pST and Heschel's gyrus, which points towards a speech-specialized role regulating the activity of coarse auditory decoding (HG) or voice-related (pST) regions. We could speculate that the right pSTS would have a similar influence upon the right micro-circuit as well, yet bilateral mirroring of the

micro-circuits functionality is limited, as we see in the rest of our results for mST and pST for instance.

To conclude on this matter, in light of recent research results, we assume the right pSTS might only contribute to the microcircuit when additional sensory information is available, meaning that the role of right pSTS is more bounded to context-dependent activity which the stimuli presented in this study were not able to activate enough.

- L. 238: “especially if one assumes that voice processing needs to happen at a prior stage to speech processing”; As I understand theories such as the AST (Poeppel, 2003), intonation and acoustic phoneme processing can occur in parallel. A possible limitation here seems that the contrast speech vs. no-voice was used as basis for the modeling. It is possible that to some extent speech-specific processing that does not reflect voice aspects is reflected in the results; (the authors also show the contrast speech vs. non-speech voice, with similar albeit less strong activations)

Response: We thank the reviewer for this comment. Indeed, to account for the different acoustic processing happening during voice and speech perception, we decided to build the DCM model based on the peak activations for both contrasts [voice > non-voice] and [speech > nonverbal]. With this choice, we aimed to ensure that the main vocal and speech features are included into the results.

It should be noted, that in the legend of Fig. 2 the wrong contrast was mentioned. Instead of [speech > non-voice] we used [speech > nonverbal/non-speech]. This might have led to confusion about which contrasts were used for the definition of the activation peaks and has been corrected accordingly.

Lastly, concerning the part of the discussion mentioned in the above comment, we adapted the section with regards to some precisions. We aimed with this study to investigate the true contrast of speech vs. voice and voice vs. other sounds processing in micro-circuits in the AC. And indeed, our observations seem to better fit a view involving more complex voice and speech processing (i.e. with parallel treatment of the information) than the commonly assumed hierarchical manner.

- I. 650: I assume the contrast is speech vs. non-verbal but voice material, could you clarify in the legend?

Response: Was changed accordingly.

- Discussion: I. 245-248: this seems to refer to the modulations Fig. 3B, which seem similar for aST in the left and the right. Why is the disintegration for the right aST emphasized, while some other modulations seem to change more? The stronger interaction between voice and speech areas in the left hemisphere seems particularly due to the additional speech node pSTS?

Response: We thank the reviewer for this comment. Our observations regarding the aST are indeed valid bilaterally. Therefore, the discussion regarding this aspect was adapted accordingly. We now write on p13 of the manuscript:

P13: “Our data overall point to some commonalities in the local AC micro-network for voice and speech processing, but they also point to considerable differences. One commonality might concern the functional contribution of the aST to the neural network architecture. The aST seems like a largely disintegrated node for voice processing bilaterally (Fig. 3b), such that voice processing in right and left AC is primarily based on an initial object classification (aST) that is then confirmed by further acoustic analysis (HG), which is rather reverse to classical hierarchical processing in AC ². In terms of the network differences, we found the pSTS as a relevant neural network node only in the left hemisphere. Regarding the role of the pSTS in the left hemisphere, it is likely that the modulation of connections between mST-pST-pSTS are more consistent than in the right hemisphere potentially due to this additional pSTS node. However, the differences between the two hemispheres for pST and mST are also noteworthy. The left pST showed neural integration according to a positive reciprocal modulation of connection with the HG (Fig. 3b), while the right pST was only negatively modulated by the HG. The left mST also seems more integrated in the micro-network in the left compared to the right hemisphere. The left mST was positively modulated by HG (while negative in the right hemisphere), and positively modulated the pST in turn (Fig. 3b). The left mST is differentially connected to HG in an effective manner when comparing the two hemispheres (Fig. 3a). Hence, besides the assumed influence of an additional node in the left hemisphere – contributing to a certain extent to a local cohesion – the bilateral micro-networks seem different when we consider their individual neural nodes and connections”.

- In the discussion: the interpretation of the effective connectivity and modulation of connection results could at times be more clearly distinguished (i.e. what part of the discussion is referring to which of the findings)

Response: We thank the reviewer for this comment. For a better clarity and at many occasions, we now link the discussion to the corresponding part in the results section, especially for data related to the GLM approach as well as for data related to the DCM approach. We hope that the readers now more clearly can link the results with the discussion section.

- Figure 3: flipping the right or left hemisphere, might make it easier to compare the network structures?

Response: We thank the reviewer for this comment. However, we believe that this is not necessary for the readability of the figure and we prefer to leave the organization left/right to avoid confusion.

Wording

- L. 56-60: the sentence structure is quite complicated, could you simplify for readability? Same in line 66-69

Response: Was changed accordingly.

- L. 88 “sound”

Response: Was changed accordingly.

References

1. Holmes, E., Zeidman, P., Friston, K. J. & Griffiths, T. D. Difficulties with Speech-in-Noise Perception Related to Fundamental Grouping Processes in Auditory Cortex. *Cereb. Cortex* **31**, 1582–1596 (2021).
2. Kumar, S., Stephan, K. E., Warren, J. D., Friston, K. J. & Griffiths, T. D. Hierarchical processing of auditory objects in humans. *PLoS Comput. Biol.* **3**, 0977–0985 (2007).
3. Aglieri, V., Chaminade, T., Takerkart, S. & P, B. Functional connectivity within the voice perception network and its behavioural relevance. *Neuroimage* **183**, 356–365 (2018).
4. Chennu, S. *et al.* Silent Expectations: Dynamic Causal Modeling of Cortical Prediction and Attention to Sounds That Weren't. *J. Neurosci.* **36**, 8305–8316 (2016).
5. Rauschecker, J. P. & Scott, S. K. Maps and streams in the auditory cortex: Nonhuman primates illuminate human speech processing. *Nature Neuroscience* vol. 12 718–724 (2009).
6. Pernet, C. R. *et al.* The human voice areas: Spatial organization and inter-individual variability in temporal and extra-temporal cortices. *Neuroimage* **119**, 164–174 (2015).
7. Morosan, P. *et al.* Human primary auditory cortex: Cytoarchitectonic subdivisions and mapping into a spatial reference system. *Neuroimage* **13**, 684–701 (2001).
8. da Costa, S. *et al.* Human primary auditory cortex follows the shape of Heschl's Gyrus. *J. Neurosci.* **31**, 14067–14075 (2011).
9. Zachlod, D. *et al.* Four new cytoarchitectonic areas surrounding the primary and early auditory cortex in human brains. *Cortex* **128**, 1–21 (2020).
10. Formisano, E., De Martino, F., Bonte, M. & Goebel, R. 'Who' is saying 'what'? Brain-based decoding of human voice and speech. *Science (80-.)*. **322**, 970–973 (2008).
11. Jones, A. B., Farrall, A. J., Belin, P. & Pernet, C. R. Hemispheric association and dissociation of voice and speech information processing in stroke. *Cortex* **71**, 232–239 (2015).
12. Venezia, J. H. *et al.* Auditory, visual and audiovisual speech processing streams in superior temporal sulcus. *Front. Hum. Neurosci.* **11**, 174 (2017).
13. Kreifelts, B., Ethofer, T., Grodd, W., Erb, M. & Wildgruber, D. Audiovisual integration of emotional signals in voice and face: An event-related fMRI study. *Neuroimage* **37**, 1445–1456 (2007).
14. Blank, H., Anwander, A. & von Kriegstein, K. Direct structural connections between voice- and face-recognition areas. *J. Neurosci.* **31**, 12906–12915 (2011).
15. Scurry, A. N., Huber, E., Matera, C. & Jiang, F. Increased Right Posterior STS

Recruitment Without Enhanced Directional-Tuning During Tactile Motion Processing in Early Deaf Individuals. *Front. Neurosci.* **14**, 864 (2020).

16. Sliwinska, M. & Pitcher, D. A comprehensive investigation of face recognition lateralisation in the posterior superior temporal sulcus. *J. Vis.* **18**, 1076 (2018).

Reviewers' comments:

Reviewer #1 (Remarks to the Author):

As stated in my original review, I felt this was a good study overall. The one concern I expressed was that the hypotheses didn't link clearly with the statistical analyses. I suggested resolving this by comparing the evidence for specific hypothesis-driven connectivity models, rather than performing an automatic search, where all nested models were treated as equally likely a priori. The authors defended their use of an automatic search, stating that they preferred a less constrained exploratory approach. I respect their decision on this, and here I will make an alternative suggestion for how to resolve my concern, via minor additions to the text rather than additional analyses.

On page 5 or 6, it would help enormously to broadly state what was expected from the DCM analysis for each outcome of the two experimental questions. For example, for the first experimental question, I suggest something like: "We expected that if the neural pathways for voice and speech are integrated, then voice and speech effects would co-localise on the same regions or connections in the DCM. Whereas, if there were separate pathways for voice and speech, these would be associated with distinct regions or connections." And for the second question: "We expected that if voice processing is organised hierarchically, the DCM would show XXX, and if it is not organised hierarchically, it would show YYY". Introducing coarse-grained hypotheses like this will clarify the narrative of the paper, and help the reader to interpret the results, without constraining the exploratory aims of the analysis.

That's my only substantive comment on this revision. In case it is helpful for the authors' future studies, I will also reply to their responses to my original review, which speak to some fundamental concepts in Bayesian statistics and DCM.

"In an unbiased approach, one can use a Bayes estimation approach to exhaustively search a large model space that considers any possible network architecture... This approach has the advantage of not being biased by prior knowledge and a researcher-bias for an a priori selection and limitation of the model space."

This is a common misunderstanding about the role of priors in Bayesian inference. Flat priors over models or parameters are not necessarily "unbiased". Rather, flat priors make the very strong statement that all models or settings of the parameters are equally likely a priori. Consider a simple Bayesian model with a single variable - the average height of a person - with a flat prior on this variable. This prior states that every height is equally likely - 165cm or 999cm or 9999999cm. This is a bad prior because it fails to take our knowledge into account, and as a result, inferences may be biased towards unrealistically large values. Similarly, the automatic search over models in DCM operates with the prior that all models are equally likely. If there are models in the search space that are not plausible - e.g. lacking connectivity between two auditory regions that are known to interact during language tasks - then the prior for those unrealistic models will be inflated (i.e., a bias). Therefore, strictly speaking, an automatic search is only appropriate where all reduced models are equally likely a priori. If the authors wanted to make this assumption transparent to a reader, they could add a sentence like "In order to take an exploratory approach with minimal constraints, we performed an automatic search over reduced PEB models, with the simplifying assumption that these models were all equally likely a priori."

Next:

"First, DCM-PEB modelling is as much as replicable as any other brain data analysis. It might be even more robust given that DCM-PEB works simultaneously on the single-subject and group-level, switching the Bayes estimation process repeatedly between both levels. Given this, each parameter thus is also not "... one separate hypothesis ...", but only meaningful when estimated with the other parameters."

I entirely agree that DCM-PEB is as replicable as any analysis, and that the parameters can only be interpreted in the context of the wider model. However, my suggestion was not about the

replicability of DCM results per se. Rather, it was about the replicability of inferences about individual connectivity parameters, relative to the replicability of inferences about models (where in the context of PEB, a model is a set of parameters). For example, if the experiment is repeated several times, one may expect to find small differences in the estimates of the parameters (i.e., their posterior expectations), and some of those differences may be large enough to change one's conclusions about whether a connection is switched on or off. Some protection against this may be gained by lumping parameters together into models. For example, one model could have speech modulating all ascending connections, and another model could have speech modulating descending connections (that's just an illustration – and does not pertain to this study specifically). A comparison of these two models' evidence is less likely to be effected by any small variations in parameter estimates than the specific estimates of each connection individually. In other words, grouping parameters together into models (where one model equals one hypothesis) can increase statistical power.

"Second, and relatedly, DCM-PEB is not a two-step procedure (single-subject modelling and estimation, followed by group-level estimations), but simultaneously models and estimates parameters on the subject/group level. So, there is no separate second-level analysis."

"Third, the PEB approach is used to compare models, but is not used as a DCM model itself. So, one cannot "... compare specific PEB models ...", but PEB is a procedure used in Bayesian model reduction (BMR)."

There may be some confusion here, which I think is important to resolve. The phrase Parametric Empirical Bayes (PEB) is used in a couple of different ways in the DCM literature. First, a "PEB model" is a mathematical object - a hierarchical model, typically with a differential equation model at the first level for each subject, and a general linear model (GLM) of the parameters at each subsequent level. Second, PEB is the name given to the software framework used to fit these models to empirical data. The software embodies a PEB model as a Matlab structure, with the log evidence for the entire hierarchical model stored in the field PEB.F and the posteriors in PEB.Ep and PEB.Cp.

Fitting a PEB model using the PEB framework implies a particular modelling procedure, and for a typical fMRI study like this, it is helpful to view it as a two-step procedure. First, the DCMs are fitted to the individual subjects' data. Then the posterior probability density over the parameters (DCM.Ep and DCM.Cp) as well as the free energy (DCM.F) are taken up to the group level. The DCM connectivity parameters are modelled at the group level using a GLM, with one regressor per covariate per connection (i.e., the total number of GLM parameters equals the number of covariates multiplied by the number of connections). The free energy of the overall hierarchical model is calculated by combining the individual subjects' free energies with the additional contribution from the GLM. Optionally, one can then go back and update the individual subjects' parameter estimates based on what's been learnt from the group – this is the "empirical Bayes" part – but that's not relevant for this study. Thus, when explaining the PEB procedure they have performed, the authors may find it useful to use the analogy of "first level analysis" and "second level analysis" that people are familiar with from SPM.

For completeness, I'll clarify the role of Bayesian model comparison and Bayesian model reduction in the PEB framework. Hypotheses are tested by comparing the evidence for the full GLM against "reduced" GLMs where particular mixtures of parameters are switched off (here, "switched off" means that certain parameters are fixed at zero by setting their prior expectation to zero and their prior variance to a very small number). For example, one could switch off all parameters relating to a particular connection, to test whether that connection exists, or turn off all parameters relating to a particular covariate, to test whether that covariate explains the data. This Bayesian model comparison is performed very quickly thanks to the use of Bayesian model reduction (BMR), which is an analytic procedure for computing the evidence and posteriors of reduced models given a full model, which differ only in their priors.

"Fourth, although DCM-PEB assumes that all reduced models are equally likely apriori, this changes over the course of the estimation procedure. DCM-PEB implements a Bayesian model reduction (BMR) approach where recursive estimations on and elimination of parameters become part of the estimation procedure as priors. The final model is not a "random" model out of equally

likely apriori models, but the model that fits the data the best after many recursive estimation processes.

"Fifth, the final model that fits the data the best and that we report in the paper is not single model out of "... thousands of possible hypotheses ..." but is the weighted average of all models included in the PEB approach. As we described in the methods section, the final step of PEB is using a Bayesian Model Averaging (BMA) approach, which creates the average of all models weighted by their posterior evidence. We think that this procedure is a much more sophisticated approach than choosing a limited number of models apriori, and then only reporting the winning model. We also feel that this approach reveals much more replicable and robust data than any selective modelling approach."

Absolutely. Indeed with both approaches – whether considering a few pre-specified PEB models or performing an automatic search – it is best practice to report parameters by computing a Bayesian model average (BMA) over the models. The difference in these approaches pertains only to which models included in the model space. One can either only include models that pertain to specific hypotheses, or include thousands of automatically generated models. I think both procedures have merit. Personally, I recommend the pre-specified model approach when there are clear hypotheses (which is why I recommended it here), and an automatic search where there are not clear hypotheses to enable an "exploratory" approach. Either way, the key principle is that the model space and priors are consistent with the scientific question being addressed.

I hope that feedback is helpful, and I'll finish this review with some suggestions for minor stylistic / text corrections:

Page 2 line 40 – "extension" -> "extent"

Page 4 line 67 – "were found" -> "have been found"

Page 5 lines 82-85 – some repetition here of the previous paragraph?

Page 5 line 98 – "concerned the notion if cortical voice" -> "concerned whether cortical voice"

Page 6 lines 142 and 144 – can "robust" be defined? Robust to what? If not, I suggest removing this word.

Page 7 line 156 – "extension" -> "extent"

Page 10 line 255 – "in specific" -> "specifically"

Signed

Peter Zeidman

Reviewer #2 (Remarks to the Author):

Overall the authors provided a good revision addressing most points, the paper was already solid before so I'm happy with the changes.

Remaining issues for me (although I'd accept editorial decision to overturn this)

- fig 1; 'we use bar plots' for not overcrowding ; well just use two lines - bar plot do not show the data and hide variance

<https://journals.plos.org/plosbiology/article?id=10.1371/journal.pbio.1002128>

- you still haven't defined 'reasonable request' ; list all available resources in the statement (neurovault, osf, etc) and explain 1- that raw data cannot be shared openly because of national law governance but can be access under xxx conditions - be explicit of those conditions - it is reasonable for me to ask them to be curated in BIDS, are they?

Reviewer #3 (Remarks to the Author):

The authors satisfactorily responded to all my comments. In my view the manuscript is ready for publication.

minor comment:

- l. 178: „the modulatory condition ..“ could you give an example here, to make it easier to follow?
- Typo: L. 140: „from a the“ remove „a“

Reviewer #1 (Remarks to the Author):

As stated in my original review, I felt this was a good study overall. The one concern I expressed was that the hypotheses didn't link clearly with the statistical analyses. I suggested resolving this by comparing the evidence for specific hypothesis-driven connectivity models, rather than performing an automatic search, where all nested models were treated as equally likely a priori. The authors defended their use of an automatic search, stating that they preferred a less constrained exploratory approach. I respect their decision on this, and here I will make an alternative suggestion for how to resolve my concern, via minor additions to the text rather than additional analyses.

Response: We thank the reviewer (Peter Zeidman) for this overall positive evaluation of our revised manuscript, and really appreciate the insightful explanations provided here in the extended comments of the reviewer to our responses to the reviewer's previous comments. In the second round of revisions, we now carefully revised our manuscript accruing the suggestions and comments mentioned below.

On page 5 or 6, it would help enormously to broadly state what was expected from the DCM analysis for each outcome of the two experimental questions. For example, for the first experimental question, I suggest something like: "We expected that if the neural pathways for voice and speech are integrated, then voice and speech effects would co-localise on the same regions or connections in the DCM. Whereas, if there were separate pathways for voice and speech, these would be associated with distinct regions or connections." And for the second question: "We expected that if voice processing is organised hierarchically, the DCM would show XXX, and if it is not organised hierarchically, it would show YYY". Introducing coarse-grained hypotheses like this will clarify the narrative of the paper, and help the reader to interpret the results, without constraining the exploratory aims of the analysis.

Response: Thanks for this suggestion(s), which we absolutely agree would make the intention of the study and the analyses much clearer. We now accordingly edited the manuscript on p5:

"More specifically, we expected that if the neural processing of voice and speech signals is integrated, activation patterns observed for these two types of voice signals would co-localize in similar brain areas of higher-order AC since separate studies have shown similar peak activity locations for voice processing ^{1,16} and for speech recognition ^{17,18} from anterior to posterior ST. However, there are also indications of a spatial separation for the neural processing of voice and speech signals ¹⁹, pointing to some functional dis-integration of both processes. The same reasoning would apply to the expected neural network underlying voice and speech processing in terms of neural (dis-)integration. If speech processing (speech as a specific voice signal) would depend on the more general voice processing (voice signals as a general category), we would expect that neural speech processing nodes would integrate with and hierarchically follow the neural voice processing nodes in terms of the neural network architecture ²⁰. However, voice

processing could also be neurally disintegrated from neural speech processing, since voice signals are also used to decode socially relevant information apart from speech information ^{21,22}”.

And on p6:

“Concerning this question of a (non-)hierarchical organization of the AC micro-networks for voice and speech processing, we expected that the neural network would follow a processing stream from primary/secondary AC to mST, and from mST to either aST (ventral stream) or pST (dorsal stream) in case of a strong hierarchical organization. In case of a non-hierarchical organization, we especially expected feed-forward and/or backward projections between low- and higher-order AC as well as a neural co-dependence of voice and speech processing (i.e. neural nodes for voice and speech processing influence each other) rather than a strict neural hierarchy (i.e. neural speech processing is dependent on voice processing nodes)”.

That’s my only substantive comment on this revision. In case it is helpful for the authors’ future studies, I will also reply to their responses to my original review, which speak to some fundamental concepts in Bayesian statistics and DCM.

Response: Very helpful indeed and clarifies many things concerning DCM and PEB.

“In an unbiased approach, one can use a Bayes estimation approach to exhaustively search a large model space that considers any possible network architecture... This approach has the advantage of not being biased by prior knowledge and a researcher-bias for an a priori selection and limitation of the model space.”

This is a common misunderstanding about the role of priors in Bayesian inference. Flat priors over models or parameters are not necessarily “unbiased”. Rather, flat priors make the very strong statement that all models or settings of the parameters are equally likely a priori. Consider a simple Bayesian model with a single variable - the average height of a person – with a flat prior on this variable. This prior states that every height is equally likely – 165cm or 999cm or 9999999cm. This is a bad prior because it fails to take our knowledge into account, and as a result, inferences may be biased towards unrealistically large values. Similarly, the automatic search over models in DCM operates with the prior that all models are equally likely. If there are models in the search space that are not plausible – e.g. lacking connectivity between two auditory regions that are known to interact during language tasks - then the prior for those unrealistic models will be inflated (i.e., a bias). Therefore, strictly speaking, an automatic search is only appropriate where all reduced models are equally likely a priori. If the authors wanted to make this assumption transparent to a reader, they could add a sentence like “In order to take an exploratory approach with minimal constraints, we performed an automatic search over reduced PEB models, with the simplifying assumption that these models were all equally likely a priori.”

Next:

“First, DCM-PEB modelling is as much as replicable as any other brain data analysis. It might be even more robust given that DCM-PEB works simultaneously on the single-subject and group-level, switching the Bayes estimation process repeatedly between both levels. Given this, each

parameter thus is also not "... one separate hypothesis ...", but only meaningful when estimated with the other parameters."

I entirely agree that DCM-PEB is as replicable as any analysis, and that the parameters can only be interpreted in the context of the wider model. However, my suggestion was not about the replicability of DCM results per se. Rather, it was about the replicability of inferences about individual connectivity parameters, relative to the replicability of inferences about models (where in the context of PEB, a model is a set of parameters). For example, if the experiment is repeated several times, one may expect to find small differences in the estimates of the parameters (i.e., their posterior expectations), and some of those differences may be large enough to change one's conclusions about whether a connection is switched on or off. Some protection against this may be gained by lumping parameters together into models. For example, one model could have speech modulating all ascending connections, and another model could have speech modulating descending connections (that's just an illustration – and does not pertain to this study specifically). A comparison of these two models' evidence is less likely to be effected by any small variations in parameter estimates than the specific estimates of each connection individually. In other words, grouping parameters together into models (where one model equals one hypothesis) can increase statistical power.

"Second, and relatedly, DCM-PEB is not a two-step procedure (single-subject modelling and estimation, followed by group-level estimations), but simultaneously models and estimates parameters on the subject/group level. So, there is no separate second-level analysis."
"Third, the PEB approach is used to compare models, but is not used as a DCM model itself. So, one cannot "... compare specific PEB models ...", but PEB is a procedure used in Bayesian model reduction (BMR)."

There may be some confusion here, which I think is important to resolve. The phrase Parametric Empirical Bayes (PEB) is used in a couple of different ways in the DCM literature. First, a "PEB model" is a mathematical object - a hierarchical model, typically with a differential equation model at the first level for each subject, and a general linear model (GLM) of the parameters at each subsequent level. Second, PEB is the name given to the software framework used to fit these models to empirical data. The software embodies a PEB model as a Matlab structure, with the log evidence for the entire hierarchical model stored in the field PEB.F and the posteriors in PEB.Ep and PEB.Cp.

Fitting a PEB model using the PEB framework implies a particular modelling procedure, and for a typical fMRI study like this, it is helpful to view it as a two-step procedure. First, the DCMs are fitted to the individual subjects' data. Then the posterior probability density over the parameters (DCM.Ep and DCM.Cp) as well as the free energy (DCM.F) are taken up to the group level. The DCM connectivity parameters are modelled at the group level using a GLM, with one regressor per covariate per connection (i.e., the total number of GLM parameters equals the number of covariates multiplied by the number of connections). The free energy of the overall hierarchical model is calculated by combining the individual subjects' free energies with the additional contribution from the GLM. Optionally, one can then go back and update the individual subjects' parameter estimates based on what's been learnt from the group – this is the "empirical Bayes" part – but that's not relevant for this study. Thus, when explaining the PEB procedure they have performed, the authors may find it useful to use the analogy of "first level analysis" and "second level analysis" that people are familiar with from SPM.

For completeness, I'll clarify the role of Bayesian model comparison and Bayesian model reduction in the PEB framework. Hypotheses are tested by comparing the evidence for the full GLM against "reduced" GLMs where particular mixtures of parameters are switched off (here, "switched off" means that certain parameters are fixed at zero by setting their prior expectation to zero and their prior variance to a very small number). For example, one could switch off all parameters relating to a particular connection, to test whether that connection exists, or turn off all parameters relating to a particular covariate, to test whether that covariate explains the data. This Bayesian model comparison is performed very quickly thanks to the use of Bayesian model reduction (BMR), which is an analytic procedure for computing the evidence and posteriors of reduced models given a full model, which differ only in their priors.

"Fourth, although DCM-PEB assumes that all reduced models are equally likely apriori, this changes over the course of the estimation procedure. DCM-PEB implements a Bayesian model reduction (BMR) approach where recursive estimations on and elimination of parameters become part of the estimation procedure as priors. The final model is not a "random" model out of equally likely apriori models, but the model that fits the data the best after many recursive estimation processes.

"Fifth, the final model that fits the data the best and that we report in the paper is not single model out of "... thousands of possible hypotheses ..." but is the weighted average of all models included in the PEB approach. As we described in the methods section, the final step of PEB is using a Bayesian Model Averaging (BMA) approach, which creates the average of all models weighted by their posterior evidence. We think that this procedure is a much more sophisticated approach than choosing a limited number of models apriori, and then only reporting the winning model. We also feel that this approach reveals much more replicable and robust data than any selective modelling approach."

Absolutely. Indeed with both approaches – whether considering a few pre-specified PEB models or performing an automatic search – it is best practice to report parameters by computing a Bayesian model average (BMA) over the models. The difference in these approaches pertains only to which models included in the model space. One can either only include models that pertain to specific hypotheses, or include thousands of automatically generated models. I think both procedures have merit. Personally, I recommend the pre-specified model approach when there are clear hypotheses (which is why I recommended it here), and an automatic search where there are not clear hypotheses to enable an "exploratory" approach. Either way, the key principle is that the model space and priors are consistent with the scientific question being addressed.

Response: Yes, very helpful, thanks for providing those very clear additional explanations.

I hope that feedback is helpful, and I'll finish this review with some suggestions for minor stylistic / text corrections:

Page 2 line 40 – "extension" -> "extent"

Response: Was changed accordingly.

Page 4 line 67 – "were found" -> "have been found"

Response: Was changed accordingly.

Page 5 lines 82-85 – some repetition here of the previous paragraph?

Response: We changed this section to avoid too much repetition from the previous paragraph. It now reads like this, p4:

“Given these previous reports of a functional homogeneity for voice processing in the TVA, including the frequently proposed voice selectivity ¹⁴, the notion of a spatially extended organization of bilateral TVAs including multiple voice subpatches seems a little bit surprising. Considering the differential working principles of AC subregions underlying the TVA ¹⁵, this uniformity might be rather unlikely”.

Page 5 line 98 – “concerned the notion if cortical voice” -> “concerned whether cortical voice”

Response: Was changed accordingly.

Page 6 lines 142 and 144 – can “robust” be defined? Robust to what? If not, I suggest removing this word.

Response: Was changed to avoid the term “robust”.

Page 7 line 156 – “extension” -> “extent”

Response: Was changed accordingly.

Page 10 line 255 – “in specific” -> “specifically”

Response: Was changed accordingly.

Signed
Peter Zeidman

Reviewer #2 (Remarks to the Author):

Overall the authors provided a good revision addressing most points, the paper was already solid before so I'm happy with the changes.

Response: We thank the reviewer again for this positive evaluation of our manuscript.

Remaining issues for me (although I'd accept editorial decision to overturn this)

- fig 1; 'we use bar plots' for not overcrowding; well just use two lines - bar plot do not show the data and hide variance

<https://journals.plos.org/plosbiology/article?id=10.1371/journal.pbio.1002128>

Response: We now included violin plots in Fig. 1d to include a representation of the data distribution.

- you still haven't defined 'reasonable request'; list all available resources in the statement (neurovault, osf, etc) and explain 1- that raw data cannot be shared openly because of national law governance but can be access under xxx conditions - be explicit of those conditions - it is reasonable for me to ask them to be curated in BIDS, are they?

Response: On p23 we now write:

“The unthresholded SPMs of all contrasts displayed in Fig. 1 were uploaded to NeuroVault: <https://identifiers.org/neurovault.collection:9707>. The DCM data and code are available on OSF: <https://osf.io/8t3aw/>. The ethical approval for this study and legal restrictions in Switzerland do not allow us to share raw data openly. The raw data that support the findings of this study could be made available from the corresponding author upon a reasonable request and in consultation with the ethical committee of the Canton Geneva (Switzerland). The sound material used in this study for auditory stimulation was included and provided by Capilla and colleagues⁵⁹.”

This statement should now include the relevant background for the restriction concerning sharing our data; this statement has also been approved now by the journal editor. Given the legal restrictions in Switzerland at the time of data collection, there is also no possibility to curate the data in BIDS.

Reviewer #3 (Remarks to the Author):

The authors satisfactorily responded to all my comments. In my view the manuscript is ready for publication.

Response: We thank the reviewer again for this positive evaluation of our manuscript.

minor comment:

- l. 178: „the modulatory condition ..“ could you give an example here, to make it easier to follow?

Response: We included an example here, p8:

“... connection (e.g. the connection from left mST to aST was allowed to be modulated by the “speech” condition, as mST was responding with higher activity to speech sounds)”.

- Typo: L. 140: „from a the“ remove „a“

Response: Was changed accordingly.

References

1. Pernet, C. R. *et al.* The human voice areas: Spatial organization and inter-individual variability in temporal and extra-temporal cortices. *Neuroimage* **119**, 164–174 (2015).
2. Staib, M. & Frühholz, S. Cortical voice processing is grounded in elementary sound analyses for vocalization relevant sound patterns. *Prog. Neurobiol.* (2020).
3. Scott, S. K., Catrin Blank, C., Rosen, S. & Wise, R. J. S. Identification of a pathway for intelligible speech in the left temporal lobe. *Brain* **123**, 2400–2406 (2000).
4. Evans, S. & Davis, M. H. Hierarchical organization of auditory and motor representations in speech perception: Evidence from searchlight similarity analysis. *Cereb. Cortex* **25**, 4772–4788 (2015).
5. Jones, A. B., Farrall, A. J., Belin, P. & Pernet, C. R. Hemispheric association and dissociation of voice and speech information processing in stroke. *Cortex* **71**, 232–239 (2015).
6. Kumar, S., Stephan, K. E., Warren, J. D., Friston, K. J. & Griffiths, T. D. Hierarchical processing of auditory objects in humans. *PLoS Comput. Biol.* **3**, 0977–0985 (2007).
7. Frühholz, S. & Schweinberger, S. R. Nonverbal auditory communication – Evidence for integrated neural systems for voice signal production and perception. *Prog. Neurobiol.* (2020). doi:10.1016/j.pneurobio.2020.101948
8. Dietziker, J., Staib, M. & Frühholz, S. Neural competition between concurrent speech production and other speech perception. *Neuroimage* **228**, (2021).
9. Yovel, G. & Belin, P. A unified coding strategy for processing faces and voices. *Trends in Cognitive Sciences* **17**, 263–271 (2013).
10. Leaver, A. M. & Rauschecker, J. P. Cortical representation of natural complex sounds: Effects of acoustic features and auditory object category. *J. Neurosci.* **30**, 7604–7612 (2010).
11. Capilla, A., Belin, P. & Gross, J. The early spatio-temporal correlates and task independence of cerebral voice processing studied with MEG. *Cereb. Cortex* **23**, 1388–1395 (2013).